# AIRS-Bench:
# A Suite of Tasks for Frontier AI Research Science Agents

## Abstract

LLM agents hold significant promise for advancing scientific research. To accelerate this progress, we introduce AIRS-BENCH (the *AI Research Science Benchmark*), a suite of 20 tasks sourced from state-of-the-art machine learning papers. These tasks span diverse domains, including language modeling, mathematics, bioinformatics, and time series forecasting. AIRS-BENCH tasks assess agentic capabilities over the full research lifecycle—including idea generation, experiment analysis and iterative refinement—without providing baseline code. The AIRS-BENCH task format is versatile, enabling easy integration of new tasks and rigorous comparison across different agentic frameworks. We establish baselines using frontier models paired with both sequential and parallel scaffolds. Our results show that agents exceed human SOTA in seven tasks but fail to match it in thirteen others. Even when agents surpass human benchmarks, they do not reach the theoretical performance ceiling for the underlying tasks. These findings indicate that AIRS-BENCH is far from saturated and offers substantial room for improvement. We open-source the AIRS-BENCH task definitions and evaluation code to catalyze further development in autonomous scientific research.

## 1 Introduction

From the rise of deep learning through to the current era of Large Language models (LLMs), the rapid progress in machine learning (ML) has been largely benchmark-driven, making robust evaluations a key requirement (Hardt, 2025). This trend is reinforced by the current *reviewing crisis*, which highlights the limits of human judgment in assessing the quality of research contributions (Xu et al., 2022; Lawrence, 2022). Recent improvements in LLM technology have extended their abilities from simple tasks to complex agentic workflows, including scientific reasoning and coding (Khatri et al., 2025; Andrews et al., 2025; Lu et al., 2024; Yamada et al., 2025). Whilst early LLM research focused on improving the capabilities of LLMs through better data and pre-training, recent work has focused on extending their abilities by leveraging more test-time compute (Yao et al., 2023a), in particular by leveraging *scaffolds* that iteratively query the LLM at test-time so as to obtain a better response. Such scaffolds can enable a test-time search of the solution space by incorporating environment feedback (Nathani et al., 2025; Toledo et al., 2025) and yield much more performant solutions. Despite this potential, we lack a standardized framework to measure how well these agents perform the actual work of a research scientist. To fill this gap, we introduce AIRS-BENCH.

We adopt the broadly accepted definition of an agent as a *computer system situated in an environment that is capable of autonomous action in order to meet its design objectives* (Wooldridge and Jennings, 1995). In our context, an *agent* is characterized as an *LLM* augmented by a *scaffold*. The LLM is the underlying probabilistic model that serves as the agent's primary reasoning core, either self-hosted OSS model or API-based. The scaffold acts as an orchestrating layer, providing the necessary mechanisms to translate LLM outputs into systematic exploration of the solution space. The scaffold is typically implemented by a *harness*, the execution framework responsible for instantiating and managing diverse scaffolding configurations. Examples are the ReAct scaffold (Yao et al., 2023b), implemented within the MLGYM harness (Nathani et al., 2025) or the Monte Carlo Tree Search (MCTS) scaffold (Kocsis and Szepesvari, 2006), implemented within the AIRA-DOJO harness (Toledo et al., 2025). We experiment with both sequential (linear) and parallel scaffolds. Sequential scaffolds follow a linear execution loop, where each LLM query is conditioned on feedback

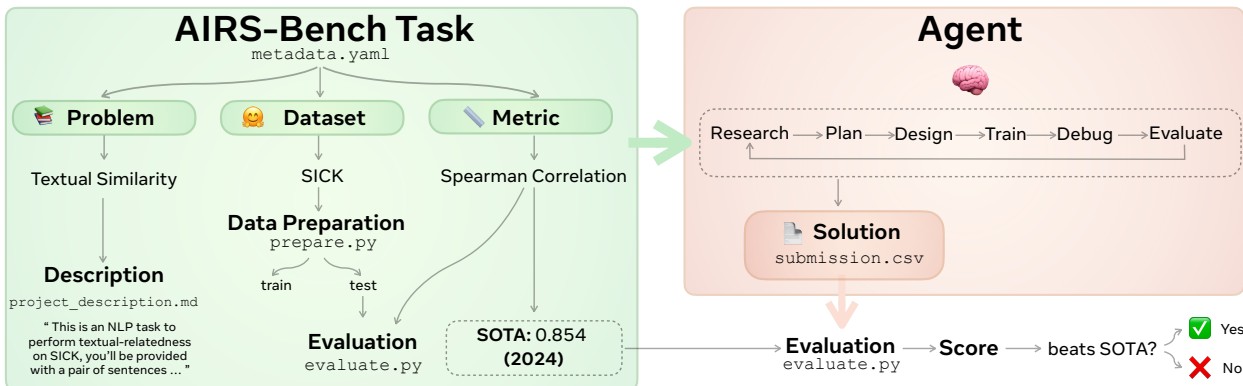

Figure 1: Example of an AIRS-BENCH task. Each task is specified by a {problem, dataset, metric} triplet. The **problem** defines the core computational challenge to be solved (e.g. textual similarity); the **dataset** specifies which data to solve the challenge over (e.g. SICK); finally, the **metric** is used to quantify performance (e.g. Spearman correlation). The agent receives the full task specification and is expected to develop a solution that generates predictions on the test labels file, which are then evaluated and compared with the state-of-the-art result.

from previous actions (Yao et al., 2023b; Nathani et al., 2025). Parallel scaffolds, by contrast, maintain and grow a population of potential solutions, utilizing data structures such as trees to guide exploration (Novikov et al., 2025; Sharma, 2025; Toledo et al., 2025).

A growing area of research for LLM agents is represented by the automation of AI research itself. We call agents designed to automate and accelerate AI research *AI Research Agents*. Accurately evaluating the performance of such agents, however, remains a significant challenge. ML has long struggled with reproducibility and statistical noise (Recht et al., 2018; Hardt, 2025), and the agentic paradigm adds further complications:

- Data contamination: LLMs are trained on vast amounts of internet data, and they often memorize benchmark solutions. This makes it difficult to assess whether an agent is truly "reasoning" on a task.

- Environmental standardization: agentic environments are difficult to standardize across different studies, and it is often unclear if success comes from the agent's capabilities or the specific way the environment was built.

- Computational cost: The high computational overhead of each autonomous run makes it difficult to conduct the extensive trials necessary to obtain statistically significant results.

These factors have exacerbated the *evaluation crisis* of AI Research Agents, as performance on existing benchmarks is increasingly obscured by pretraining leakage, inconsistent environment setups, and noisy empirical evaluations (Haimes et al., 2024; Dehghani et al., 2021). We introduce the *AI Research Science Benchmark* (AIRS-BENCH) to address these limitations and provide a standardized evaluation of AI Research Agents. The philosophy behind AIRS-BENCH is that, much like the progress in ML, the advancement of AI Research Agents should be a benchmark-driven process. AIRS-BENCH is a suite of 20 agentic tasks, curated from recent state-of-the-art (SOTA) literature. This ensures that tasks are both challenging and relevant to the broader ML community. In our design, agents are not required to tackle these tasks through direct inference. Instead, they must generate the code necessary to train and validate an ML model. We then evaluate the agent's capabilities by executing the code it generated and measuring the model performance. This approach effectively evaluates the agent's capacity to function as an autonomous research scientist.

To both standardize evaluations and make the tasks closely match their original versions in the ML literature, we developed a *task configuration standard* that encapsulates an AIRS-BENCH task (see Figure 1 for an

example) and a task creation pipeline based on semi-manual sourcing, creation, reviewing and verification of tasks. Our proposed AIRS-Bench task standard can be adapted and extended to virtually any ML problem, effectively democratizing AI agentic research. For fair evaluations across all agents we have designed our experiments to make runs as comparable as possible by factoring in infrastructure issues (e.g. crashing runs) and eliminating environment discrepancies. To investigate the impact of different scaffolds, we evaluate both the AIRA-dojo (Toledo et al., 2025) and MLGym (Nathani et al., 2025) harnesses combined with different LLMs (CWM (Carbonneaux et al., 2025), GPT-4o (OpenAI, 2024a), gpt-oss-20b and gpt-oss-120b (OpenAI, 2024b), o3-mini (OpenAI, 2025b) and o3 (OpenAI, 2025c), GPT-5 (OpenAI, 2025a) and GPT-5.4 (OpenAI, 2026), Claude Opus 4.5 (Anthropic, 2025) and Opus 4.6 (Anthropic, 2026), Gemini 3.1 Pro (Google DeepMind, 2025), Muse Spark (Meta AI, 2026) and Devstral (Rastogi et al., 2025)).

We also introduce an evaluation protocol, consisting of different metrics and aggregating results across seeds and tasks to allow for statistically robust and interpretable results. The key contributions of the paper are:

- **Uncontaminated benchmark for AI Research Agents**: AIRS-Bench uniquely assesses agents on the end-to-end ML research workflow: idea generation, methodology design, experiment analysis, and iterative refinement—without access to baseline code. This enables a realistic evaluation of agentic research abilities. AIRS-Bench tasks cover a variety of machine learning research problems, ranging from NLP, math and code to biochemical modeling and time series forecasting. The benchmark is designed to be compatible with multiple agentic frameworks (i.e. harnesses), supporting robust and fair comparisons across different agent architectures and tool integrations.

- **Task configuration standard**: we introduce a task configuration standard and fixed evaluation metrics, to ensure reproducibility and minimize runtime and environment inconsistencies. We establish quality by leveraging human checks throughout task building, reviewing, and verification.

- **Empirical analysis**: we benchmark all tasks across frontier open and closed-source models and different scaffolds, revealing substantial variation in performance. We closely inspect cases where agents are shown to match or beat human SOTA and analyze the agent-generated solutions—including cases where agents discover effective SOTA-exceeding combinations of approaches.

- **Evaluation protocol**: we define a suite of metrics assessing different aspects of agents' capabilities including valid submission rates, normalized performance scores, and Elo ratings. Normalization enables us to aggregate performance across tasks by applying transforms that map 0.0 scores to the weakest valid solution and 1.0 to human SOTA. The metrics used are designed to accurately reflect progress on developing state-of-the-art AI Research Agents. We report results on these metrics across the range of selected agents; our results suggest that the benchmark is far from solved, leaving plenty of headroom for development of AI Research Agents.

The rest of the paper is organized as follows. Section 2 contains an overview of existing benchmarks within the AI research domain. Section 3 introduces the agent, scaffold and harness definitions and section 4 presents the benchmark structure and creation methodology. Sections 5 and 6 outline our experimental design and results. We conclude with discussing learnings and key findings in Section 7.

## 2 Related Work

We examined a number of benchmarks evaluating the ability of AI agents to carry out scientific research, including ML-related agentic tasks. Below, we outline several trends we identified and how they relate with AIRS-Bench.

**Full cycle of Scientific Method**. Many earlier papers create agentic tasks from available sources: Github repositories (CSR-bench; Xiao et al., 2025), top-tier conference papers and their data (PaperBench; Starace et al., 2025a), Kaggle competitions (MLE-bench; Chan et al., DSBench; Jing et al., 2025), ML research competitions (MLRC-Bench; Zhang et al., 2025), and carefully selected cross-domain papers (SciReplicate-bench; Xiang et al., 2025b). The experimental cycle of these benchmarks can include ideation, implementation, experimentation, analysis and comparison to previous results, with several benchmarks focusing on

Table 1: Comparison of 14 popular agentic AI research benchmarks with AIRS-Bench across key evaluation dimensions. AI Research data denotes whether benchmark was built with data sourced from AI Research literature. Task horizon refers to the time window provided to solve the problem and it can be short (<1 hour), medium (1-12 hours) or long (>12 hours). We indicate whether the benchmarks assess for the hypothesis generation (**H**), implementation (**I**), experimentation (**E**) and analysis (**A**) stages of the scientific pipeline. Compute refers to the resources provided to the agent and it can be CPU, low GPU ($\leq$ 1 hour per task) and high GPU (>1 hour per task). No baseline refers to whether the agent has access to a baseline solution to tackle the problem.

| Benchmark | Task Composition and Origin | AI Research Data | Task Horizon | Scientific Method (H) (I) (E) (A) | | | | No Baseline | Agent Compute |
|---|---|---|---|---|---|---|---|---|---|
| **AIRS-Bench (ours)** | 20 tasks from 17 machine learning papers with state-of-the-art results | ✓ | long | ✓ | ✓ | ✓ | ✓ | ✓ | high GPU |
| Automated LLM Speedrun (Zhao et al., 2025) | 76 tasks from the NanoGPT Speedrun Github repo | ✓ | long | x | ✓ | ✓ | ✓ | ✓ | high GPU |
| Auto-Bench (Chen et al., 2025a) | 6 graph discovery tasks from chemistry and social networks | x | short | ✓ | x | x | x | ✓ | not specified |
| CORE-Bench (Siegel et al., 2024a) | 270 tasks from 90 social, medical, and computer science papers | ✓ | medium | x | ✓ | ✓ | ✓ | x | low GPU |
| CSR-Bench (Xiao et al., 2025) | 107 Github repos from CV, NLP and interdisciplinary ML papers | ✓ | not specified | x | ✓ | ✓ | ✓ | x | not specified |
| MLE-Bench (Chan et al., 2024) | 75 Kaggle competitions | x | long | ✓ | ✓ | ✓ | ✓ | ✓ | high GPU |
| MLGym-Bench (Nathani et al., 2025) | 13 tasks from supervised learning, RL and algorithmic reasoning problems | ✓ | medium | ✓ | ✓ | ✓ | ✓ | ✓ | high GPU |
| ML-Agent-Bench (Tang et al., 2024) | 13 tasks from Kaggle, computer science papers, and textbooks | ✓ | medium | ✓ | ✓ | ✓ | ✓ | x | low GPU |
| SWE-Bench (Jimenez et al., 2023) | 2,294 Github issues from open-source repos | x | short | x | ✓ | ✓ | ✓ | ✓ | CPU |
| SciReplicate-Bench (Xiang et al., 2025a) | 100 tasks from 36 NLP papers | ✓ | medium | x | ✓ | ✓ | ✓ | x | not specified |
| PaperBench (Starace et al., 2025b) | 8,316 rubrics from 20 ICML papers | ✓ | short | x | ✓ | ✓ | ✓ | x | low GPU |
| ResearchBench (Liu et al., 2025a) | Hypothesis generation task for 1386 papers across the sciences | ✓ | not specified | ✓ | x | x | ✓ | ✓ | not specified |
| RE-Bench (METR, 2024) | 7 LLM pretraining/coding tasks and 71 human attempts | x | medium | ✓ | ✓ | ✓ | ✓ | x | high GPU |
| LMR-Bench (Yan et al., 2025) | 28 code reproduction tasks from 23 NLP papers | ✓ | not specified | x | ✓ | x | x | x | not specified |
| PostTrainBench (Rank et al., 2026) | Post-train 4 base LLMs to maximise perf across 7 benchmarks | ✓ | medium | ✓ | ✓ | ✓ | ✓ | ✓ | high GPU |

separate stages of the cycle, such as IdeaBench (Guo et al., 2024), LiveIdeaBench (Ruan et al., 2025), AI Idea Bench (Qiu et al., 2025) and ResearchBench (Liu et al., 2025b) for ideation, FML-bench (Zou et al., 2025) for idea novelty estimation, SurveyBench (Sun et al., 2025) for literature review, and DataGovBench (Liu et al., 2025c), DCA-Benchmark (Huang et al., 2025), DA-Code (Huang et al., 2024b), and DS-1000 (Lai et al., 2022) for data quality, implementation and analysis. AIRS-Bench, on the other hand, requires the agent to excel in every step of the scientific method to perform well on the benchmark tasks, spanning hypothesis generation, implementation, experimentation, and analysis.

**Access to Baseline Solution**. Benchmarks differ significantly in two ways: whether the agent is granted access to a baseline solution, and the degree of "saturation" within the underlying problem. AIRS-Bench prioritizes unsaturated tasks, and does not provide a starter solution to the agent. This approach makes AIRS-Bench tasks challenging, as agents must navigate a longer reasoning horizon to make progress independently.

**Different environments**. Recent benchmarks leverage a wide range of different environments. While most setups closely resemble AIRS-BENCH—requiring agents to interact via prompts and code execution—others incorporate gamified environments (DiscoveryWorld; Jansen et al., 2024) or physics engines (FE-ABench; Mudur et al., 2024). Task sets are typically manually curated, though some benchmarks (DiscoveryBench; Majumder et al., 2024, SUPER; Bogin et al., 2024) also include synthetic tasks. Some papers simulate user interactions within the environment (AppWorld; Trivedi et al., 2024).

**Diversity of domains**. Beyond core ML tasks, there is a growing trend to expand agentic evaluation into broader scientific domains, such as bioinformatics, chemistry, and physics. We identify more than 20 different scientific domains covered across recent benchmarks, including DiscoveryWorld (Jansen et al., 2024), DiscoveryBench (Majumder et al., 2024), CURIE (Cui et al., 2025), FEABench (Mudur et al., 2024), ScienceAgentBench (Chen et al., 2025b), CORE-bench (Siegel et al., 2024b), AUTObench (Chen et al., 2025a), ResearchBench (Liu et al., 2025b) and BioMLbench (Miller et al., 2025). AIRS-BENCH features text and tabular-based tasks across seven categories of ML problems, including language modeling, mathematics, code generation, molecular modeling, and time-series forecasting. Currently, AIRS-BENCH provides a comprehensive testbed for agents, with the potential for future expansion into additional scientific domains.

We selected a subset of benchmarks that most closely resemble AIRS-BENCH—specifically those evaluating an agent's ability to conduct AI research independently. Table 1 summarizes this comparison across several dimensions: task composition and origin, reasoning horizon, the stages of the scientific method covered, access to baseline solutions, and compute requirements. AIRS-BENCH is composed of tasks grounded in AI research data and crafted so that AI Research Agents (i) tackle long-horizon research challenges that mirror the complexity of scientific discovery (ii) engage with the complete scientific pipeline from hypothesis generation to implementation, experimentation and analysis in an iterative fashion (iii) operate without a starting solution to ensure unbiased evaluation and promote original research and (iv) leverage substantial computational resources to enable thorough solution space exploration and tackle modern AI research problems. These characteristics make AIRS-BENCH well-suited for tracking progress toward developing state-of-the-art AI Research Agents.

## 3 Agents, Scaffolds, Harnesses

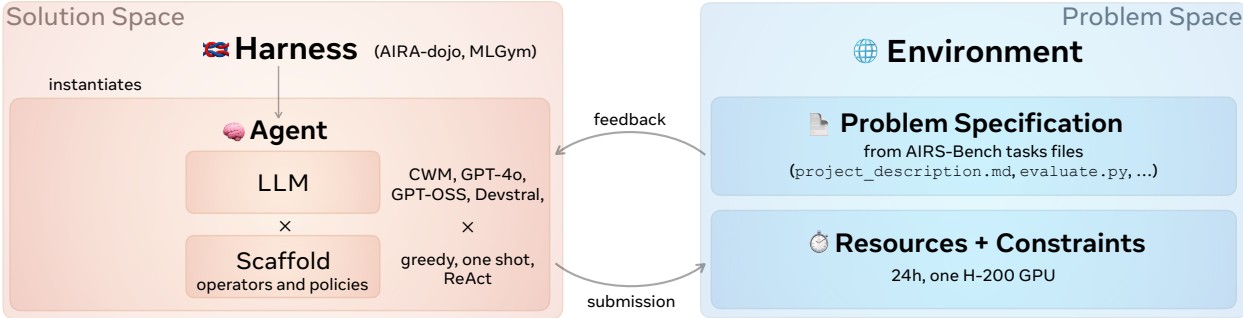

Figure 2: We define an **agent** as a pair consisting of a large language model (LLM) and a scaffold. A **scaffold** comprises a set of mechanisms, such as operators and search algorithms, that enable the LLM to explore the solution space effectively. Scaffolds are instantiated by a **harness**, which serves as a system that encapsulates the agent and manages its research process. The **environment** provides the agent with the problem specifications, as well as any constraints and resources available for its exploration.

In line with the broader agent research literature, we view an agent as a computer system that is situated in some environment and and is able to act autonomously in this environment in order to achieve its design objectives (Wooldridge and Jennings, 1995). For our specific context of LLM agents for AI Research, we define an *agent* as the combination of an LLM and a *scaffold*. A *scaffold* is an algorithm, expressed as a a set of operators and search policies governing how the agent searches the space of possible solutions for the task at hand. Scaffolds are instantiated within a *harness*—a system that wraps the agent and manages its

execution (see Figure 2). Examples of harnesses include AIDE (weco.ai, 2024; Jiang et al., 2025), Claude Code, [1] OpenCode, [2] MLGYM (Nathani et al., 2025) and AIRA-DOJO (Toledo et al., 2025). For instance, we refer to greedy search as a scaffold that is instantiated within the AIRA-DOJO harness.

**AIRA-dojo** is a harness that enables the agent to evolve its solution through a set of operators and a search policy. The search policy (e.g., greedy search, Monte Carlo Tree Search (Kocsis and Szepesvari, 2006), evolutionary algorithms (Novikov et al., 2025; Sharma, 2025; Romera-Paredes et al., 2023)) guides the exploration of the solution space, while the operators modify existing solutions to generate new candidate solutions. This process results in a tree structure, where each node represents a Python code solution, created by one of the following operators: (1) *Draft*, which generates the initial set of solutions; (2) *Debug*, which identifies and corrects errors within a given node; and (3) *Improve*, which enhances a solution to increase its performance according to specified evaluation criteria. AIRA-DOJO operators enhance AIDE to (i) promote solution diversity, by inserting into the context solutions of sibling nodes, and (ii) facilitate bug fixing, by including in the context the entire ancestral memory of the solution's debug chain. AIRA-DOJO allows access to the internet, including additional data and pretrained models, which are loaded in the cache for the agent to use according to prompting instructions. AIRA-DOJO's operator prompts can be found in Appendix D.2.

**MLGym** is a harness through which an agent can sequentially improve its initial solution in a ReAct-like manner (Yao et al., 2023b). The agent can develop a solution based on its own ideation and feedback from the execution of the implementation. Access to the internet is also allowed in a manner similar to AIRA-DOJO, with the option to cache pretrained models and instruct the agent about them in the prompt. The agent has access to tools and bash, and the full system prompt is present in Appendix D.1.

## 4 Method

**AIRS-Bench** consists of 20 tasks extracted from 17 machine learning papers and 16 different datasets. To ensure selection of highly impactful tasks for the community, we sourced tasks and SOTA results from papers published at well-known AI conferences and journals as well as arXiv preprints (see Appendix G for a distribution of publication venues and years). The initial list of tasks was sourced from PapersWithCode's leaderboards (Taylor, 2020; Kardas et al., 2020) and filtered down for datasets that had a state-of-the-art result published between 2020 and 2025; additional criteria applied at this stage included availability of the dataset (preferably via HuggingFace) and existence of a train and a test split. After verifying these requirements, we manually created each task and reviewed the SOTA result associated with it, by confirming the number reported in the cited paper, checking that the evaluation metric and dataset splits were identical and updating the SOTA result if a better score could be found. By following this methodology, we initially created and evaluated approximately 100 tasks from ∼85 different machine learning papers and datasets. We subsequently selected 20 out of these to ensure tractability and accuracy of the benchmark. The process is outlined in Appendix B.

### 4.1 Tasks Diversity

Figure 3 show the 7 distinct categories that AIRS-BENCH's 20 tasks are organized into, with the tasks spanning a broad spectrum of ML problems. The largest share is dedicated to the NLP domain, comprising *Question Answering* (4/20, open-ended and extractive question answering over diverse contexts), *Text Extraction & Matching* (3/20, e.g. coreference resolution, semantic similarity) and *Text Classification* (2/20, sentiment analysis, document categorization) tasks. Then, *Molecules and Proteins ML* tasks (5/20) focus on predicting molecular properties and solving biochemical modeling problems, whereas *Time Series* tasks (3/20) perform forecasting over temporal data. Finally, *Code* (2/20) and *Math* (1/20) tasks correspond to code generation/retrieval and mathematical reasoning, respectively.

The datasets included in AIRS-BENCH span both unstructured data (plain text) and structured modalities (such as tables and graphs), presenting challenges for even the most advanced models (Fatemi et al., 2023; Lupidi et al., 2025).

---

[1]https://code.claude.com/docs/
[2]https://github.com/anomalyco/opencode

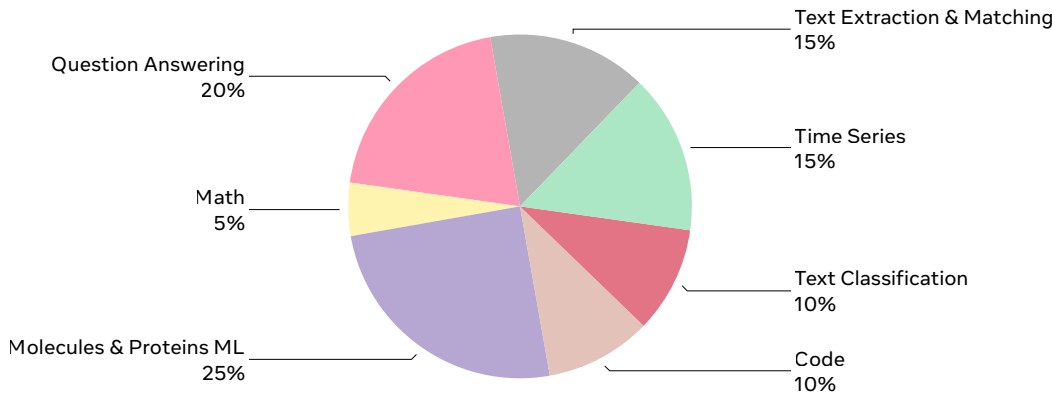

Figure 3: Distribution of AIRS-BENCH tasks by category. We consider 7 distinct task categories in total: *Code, Math, Molecules & Proteins ML, Question Answering, Text Classification, Text Extraction & Matching,* and *Time Series.*

Table 2: Core fields of the `MathQuestionAnsweringSVAMPAccuracy` task stored in its `metadata.yaml` file.

| Field | Value |
|---|---|
| name | MathQuestionAnsweringSVAMPAccuracy |
| research_problem | Math Question Answering |
| dataset | ChilleD/SVAMP |
| metric | Accuracy |
| metric_lower_is_better | false |
| config | default |
| train_split | train |
| test_split | test |
| input_columns | question_concat |
| scoring_column | Answer |
| category | Math |
| sota_score | 0.942 |
| sota_year | 2025 |
| sota_venue | Frontiers of Computer Science |
| sota_paper_title | *Achieving >97% on GSM8K: Deeply Understanding the Problems Makes LLMs Better Solvers for Math Word Problems* |
| sota_paper_url | https://arxiv.org/pdf/2404.14963v5 |
| dataset_paper_url | https://arxiv.org/abs/2103.07191 |

## 4.2 Key Task Fields

Table 2 lists the core task fields of an AIRS-BENCH task. Specifically, `name` contains the unique task name, `research_problem` describes the problem the agent is asked to solve, `dataset` contains the HuggingFace dataset identifier, `config` corresponds to the subset of the HuggingFace dataset used and `train_split` and `test_split` to the HuggingFace datasets splits used by the agent for training and evaluation, respectively. The `metric` field contains the metric the agent is asked to minimize or maximize (depending on the value of `metric_lower_is_better`), while `input_columns` contains the dataset columns the agent can utilize to solve the research problem and `scoring_column` is the column the agent's solution will be evaluated against.

Finally, `category` describes the domain of the research problem, `sota_paper_title`, `sota_paper_url`, `sota_year` and `sota_venue` hold information related to the paper containing the state-of-the-art result and `sota_score` is the state-of-the-art value of the `metric` appearing in the paper; the agent is not accessing any of the information stored in the category and sota-related fields.

### 4.3 Task Files

Table 2 describes most of the key information that forms an AIRS-Bench task. This information requires additional code and data to be run on the AIRA-dojo and MLGym scaffolds. Sections D.1 and D.2 of the Appendix contain the system prompts of the two scaffolds. The full task specification includes a folder, whose name is the same as the task name, and a number of files that are required for the agent to solve the task within AIRA-dojo. The task files are organized in such a way that they can be easily and programmatically converted into files required by different harnesses; we show this by programmatically converting them into task definition files for the MLGym agentic framework. The linked Github repository[3] contains the AIRS-Bench task specifications for AIRA-dojo and MLGym along with scripts for AIRA-dojo-to-MLGym format conversion and scripts for preparing the experimental environment (e.g. dataset downloads). In the remaining of this section, we describe the format and purpose of the task definition files for AIRA-dojo.

#### 4.3.1 project_description.md

The `project_description.md` file contains the instructions provided to the agent to complete the task in the form of a lengthy and appropriately structured prompt; see Appendix C.1.1 for a full example. The prompt is divided into three sections: a description of the research problem, a description of the dataset and an explanation of the evaluation setup.

The research problem is presented in a sentence that describes the objective of the problem along with the column of the dataset that the predictions will be evaluated against. For the running example, this is: "Your task is to solve math word problems. Each example presents a short story followed by a specific question. Your task is to read the text and predict the correct numerical answer. Your predictions will be scored against the `Answer` column of the test set "

In the dataset description, we report the structure of the HuggingFace dataset. In particular, we specify which repo the data comes from and the dataset schema (features) with an overview of the columns and their datatypes. This helps the agent understand how the data looks like even if the scaffold does not provide a lookahead function. All the data used by the agent during a task is pre-downloaded and exported to the agent's container.

Lastly, we explain to the agent how to submit the solution and how it will be scored: the agent is expected to submit its solution in the form of a `.csv` file containing the predictions on the test split, which has the benefit of a standard output format.

We also provide the agent with the code of the evaluation script (`evaluate.py`, explained below) that contains the metric implementation and will be used to score the agent-produced `submission.csv` file against the test data.

#### 4.3.2 prepare.py and evaluate_prepare.py

The `prepare.py` and `evaluate_prepare.py` files contain the one-time data preparation logic for the agent to solve the problem and for the evaluation of the agent's solution, respectively. We use two separate scripts as there are differences between the two settings; e.g. the test labels need to be removed while the agent is building its solution.

#### 4.3.3 evaluate.py

The `evaluate.py` file is the evaluation script used to score the agent's submissions against the test data. See Appendix C.1.2 for the evaluation script of the `MathQuestionAnsweringSVAMPAccuracy` task. The script contains three core functions: `load_test_set` where the script loads the test data, `evaluate` which implements the metric used to score the submissions, and `cli` which orchestrates loading of the agent's submissions and test data, running the `evaluate` method on these, and reporting the results to stdout.

---

[3]https://github.com/facebookresearch/airs-bench

### 4.3.4 metadata.yaml

The `metadata.yaml` file contains all the metadata about the task (same as the fields of Table 2 described in detail above) along with additional requirements to run the task (like libraries used by the evaluation script that need to be installed). See Appendix C.1.3 for the `metadata.yaml` file of the `MathQuestionAnsweringSVAMPAccuracy` task.

### 4.3.5 utils.py

The `utils.py` file is an optional file to consolidate overlapping code between the `prepare.py`, `evaluate.py` and `evaluate_prepare.py` files. Examples include normalization transforms used for data preparation or bespoke label extraction logic.

### 4.3.6 Train and test datasets

All datasets have been downloaded beforehand using the `prepare_hf_datasets_text.py` script. The data required for the task is then mounted to the agent's container and prepared using the code within `prepare.py` and `evaluate_prepare.py`. The data folder always contains a train split under a folder whose name is the value of the `train_split` field of `metadata.yaml` and a test split under a folder similarly named using the `test_split` field. For the train split, one preparation step is removing all but the relevant columns for the task (and transforming their content if needed). This is to ensure that the model does not have access to extra data that might hint to the solution. The test split contains the test labels when used by the evaluation script to score the agent's submissions, but it does not contain the labels when accessed by the agent to look at the structure of the test set and solve the problem.

## 5 Experiments

### 5.1 Evaluation Setup

We evaluate AIRS-BENCH using two harnesses, MLGYM and AIRA-DOJO. To isolate the effects of design choices associated with different harnesses, we ensured similar constraints and resources across all runs. For further details, refer to Table 5 in Appendix D. The greedy scaffold within AIRA-DOJO explores several solutions through a tree-based search policy, while MLGYM operates sequentially within one reasoning stream. Each run lasts for 24 hours with access to one H-200 GPU. We launch each run of each task at least 10 times (which we refer to as 10 "seeds").

Throughout AIRS-BENCH evaluations with MLGYM and AIRA-DOJO harnesses, agents are allowed to access HuggingFace checkpoints, permitting the use of pretrained models. To facilitate this process and to mitigate HuggingFace rate limits, we locally cache a number of pretrained checkpoints. Note that the cache does not offer access to the latest foundational models and the most recent cached model dates back to 2021. The full list can be found in section F of the Appendix. For both AIRA-DOJO and MLGYM runs, agents were explicitly instructed about the existence of the cache. Across all experiments, we do not provide agents with any information regarding the methodology or score of the SOTA paper. We hypothesize that some of the tasks in the benchmark would have benefited from more compute or time, but we kept costraints uniform across tasks to provide the agents with the same resources and push the limits of their ideation capabilities.

### 5.2 Metrics and Score Aggregation

We evaluated the performance of the agents on AIRS-BENCH using three metrics: mean valid submission rate, average normalized score and Elo rating. The definitions of these metrics are provided below. Throughout all metrics and empirical results, we follow the terminology introduced in Section 3, for which an agent $a$ is defined as the combination of a scaffold (e.g. Greedy) and a base LLM (e.g. gpt-oss).

For each task, the agents faced the challenge of being able to *submit* a valid solution, i.e. one that meets the requirements specified in the task description and yields a valid score. Our first evaluation metric is thus the mean **valid submission rate** (VSR) across tasks for an agent $a$, defined as

$$\overline{\text{VSR}}_a = \frac{1}{N_a} \sum_{t=1}^{N_a} \frac{valid_{a,t}}{total_{a,t}} \tag{1}$$

where $valid_{a,t}$ is the number of valid (successful) runs for agent $a$ on task $t$, $total_{a,t}$ is the number of total runs for agent $a$ on task $t$, $N_a$ is the number of tasks over which agent $a$ is being evaluated (i.e. the AIRS-BENCH tasks) and agent $a$ is a combination of a base LLM with a scaffold. The mean valid submission rate assesses the agents' capability to come up with a working solution and submit it confidently.

Producing an aggregate score for AIRS-BENCH is challenging due to the high diversity of tasks included: most tasks have unique metrics, and even for tasks sharing the same metric (e.g. accuracy), ranges reported in the literature for each of them may vary significantly. To aggregate heterogeneous metrics and ranges into a common scoring system, we define the **normalized score** (NS) of an agent $a$ on a task $t$ as:

$$\text{NS}_t^a = \frac{\phi_t(s_t^a) - \phi_t(s_t^{\min})}{\phi_t(s_t^{\text{sota}}) - \phi_t(s_t^{\min})} \tag{2}$$

where $s_t^{\min}$ corresponds to the worst score observed across all seeds and all agents on task $t$, $s_t^{\text{sota}}$ is the SOTA score on task $t$ sourced from literature, $s_t^a$ is the score achieved by agent $a$ on task $t$ and $\phi_t$ is a non-linear transformation. Note that $\phi_t(s_t^{\min})$ and $\phi_t(s_t^{\text{sota}})$ will always correspond to normalized scores of 0 and 1, respectively. Equation 2 involves a two-step normalization: first, for a given task $t$, we apply a monotonic map $\phi_t$ onto the raw score $s$ achieved by the agent's submission; second, we restrict the resulting scores within the $[0, 1]$ interval if the agent performs worse than or same as SOTA and $> 1$ if the agent exceeds SOTA to enable subsequent aggregation across tasks. Specifically, we employ the **march of 9s** transform[4] as our choice of $\phi_t$, defined as

$$\phi_t(s) = -\log_{10}(|s - s_t^{\text{opt}}|) \tag{3}$$

where $s_t^{\text{opt}}$ is the overall possible optimal score for the task (e.g., 1.0 for classification accuracy, 0.0 for regression error), as opposed to the best score obtained or SOTA (which e.g. for accuracy would be less than 1.0). This choice of $\phi_t$ is to adjust changes of the score so that they reflect intuitive measures of progress on the benchmark: this approach treats closing e.g. the gap from 0.99 to 0.999 as significant as closing the gap from 0.9 to 0.99, since both represent a tenfold reduction in the distance to optimal.[5] When averaging normalized scores across seeds, we include both failed (i.e., the agent fails to submit a valid solution) and invalid submissions (i.e., the agent submits a solution that does not yield a numerical score) treating them as submissions with 0 normalized score.

Lastly, to quantify the relative skill level of each agent evaluated, we employ the **Elo rating system** (Elo, 1967). We do so by treating each agent as a player. For each AIRS-BENCH task, we treat each pairwise comparison of the agents' scores as a game, with the agent producing a better score winning that game. If two agents do not produce a valid submission or both produce the same score, that game is considered a tie.

We estimate the agents' ratings by fitting a Bradley–Terry (BT) model to the head-to-head outcomes, following the approach used in Chatbot Arena (Chiang et al., 2024). The model infers latent skill parameters ($\theta_a$ for agent $a$) such that the probability of one agent $a$ outperforming another agent $b$ follows a logistic function of their skill difference:

$$P(a > b) = \frac{1}{1 + \exp(\theta_b - \theta_a)} \tag{4}$$

We then convert the estimated skill parameters $\theta_a$ into Elo ratings using the following transformation, where N denotes the total number of evaluated agents:

$$R_a = \frac{400}{\ln(10)} \cdot \left[ \theta_a - \frac{1}{N} \sum_{k=1}^{N} \theta_k \right] + 1000 \tag{5}$$

---

[4]The "march of nines" is a metaphorical expression, popularized by AI researcher Andrej Karpathy, to describe the vast and non-linear engineering effort required to achieve higher levels of reliability in AI systems (Karpathy and Patel, 2025).

[5]The reader can contrast this with a simpler but less representative transform definition, such as the identity transform, for which we present results in Section C of the Appendix

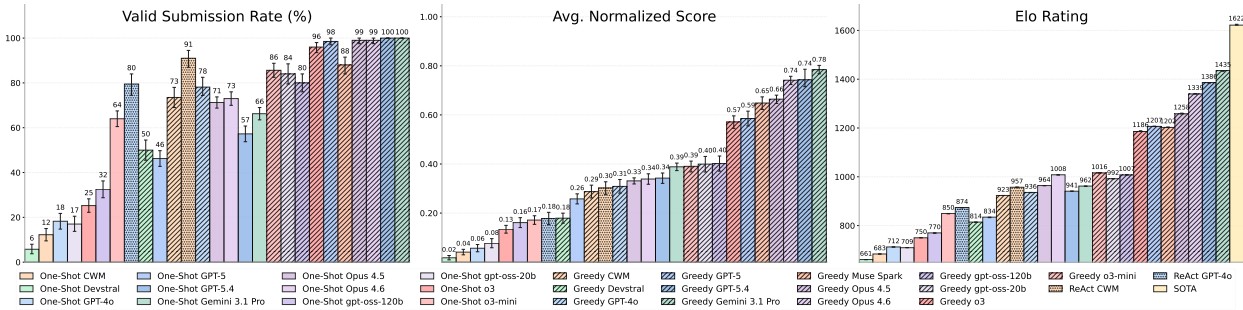

Figure 4: Overall performance of the 27 evaluated agents on the three metrics introduced in Section 5.2, namely valid submission rate, average normalized score and Elo rating. Results are ordered by increasing average normalized score.

Unlike the classical Elo calculation, the BT model is order-invariant, making it well-suited for batch evaluations across multiple agents and tasks. Please note that in our setting we treat the human SOTA scores as an additional agent, the "SOTA" agent.

## 6 Results

We evaluate a total of 27 agents, i.e. LLM-scaffold pairs. The language models used are the Code World Model (CWM), o3-mini, o3, GPT-5, GPT-5.4, gpt-oss-20b, gpt-oss-120b, GPT-4o, Devstral-Small 24B, Claude Opus 4.5, Claude Opus 4.6, Gemini 3.1 Pro and Muse Spark. We evaluate three scaffolds: (i) One-Shot, where the agent can attempt solving the problem only once with the same set of AIRA-DOJO operators (by definition that would be the *Draft* operator only) (ii) Greedy, where the agent performs greedy search with the AIRA-DOJO operator set and (iii) ReAct, where the ReAct prompting technique implemented by MLGYM is powering the agent.

### 6.1 Comparing performance across agents

Figure 4 provides an overview of the three aggregate benchmark metrics introduced above. We observe that the reasoning models (e.g. the latest frontier models Gemini 3.1 Pro, GPT-5.4, Opus 4.6) perform better in both one-shot and greedy settings, with more recent releases achieving sustained improvements (e.g. Greedy Opus 4.6 vs 4.5). Model size helps with performance, e.g. One-shot gpt-oss-120b outperforms One-shot gpt-oss-20b; however, size alone does not determine performance, with test-time scaling eventually making up for it, i.e. Greedy gpt-oss-120b is on par with Greedy gpt-oss-20b. Moreover, tree-search methods benefit agents powered by both open-source and closed-source models, as suggested by the sizeable gaps between the performances of Greedy and One-Shot variants across all model families. At the same time, performance of agents backed by linear scaffolds, such as ReAct CWM and ReAct GPT-4o, varies; in one case the ReAct agent is better than its greedy counterpart (CWM) and in the other case worse (GPT-4o). We observe that the relative ranking of agents shows similar (but not identical) trends for the average normalized score and Elo Rating performance metrics. However, the ability to submit a valid solution does not always correlate with the ability to submit a high-performing solution. This suggests that agents who are willing to take risks may achieve better performance. Finally, the majority of agent-task combinations achieve results between the worst possible score and human SOTA, with a fraction of them (7 out of 20 tasks) exceeding SOTA, primarily driven by greedy search strategies.

We present valid submission rates for each agent in Figure 5, highlighting different ranges of valid submission rate across all tasks. We consider four submission rate ranges: *invalid* indicates that the agent failed to submit a valid solution across all its seeds; *low/medium/high* correspond to the 1–33%, 34–66% and 67–100% of valid submission rates. Six agents, namely ReAct CWM, Greedy GPT-5, Greedy GPT-5.4, Greedy Opus 4.5, Greedy Opus 4.6, and Greedy Gemini 3.1, achieve a high submission rate on all 20 tasks.

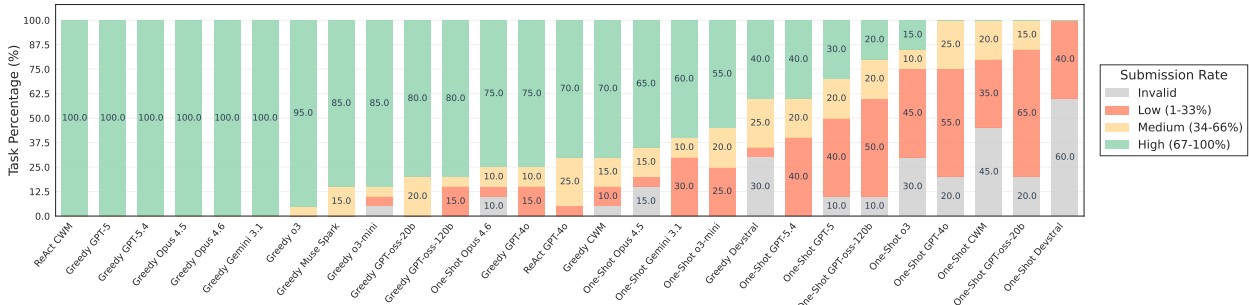

Figure 5: Submission rate distribution for the 27 agents tested. Each bar shows the distribution of submission rates across tasks for a given agent. The categories are defined as follows: *invalid* indicates that the agent did not provide any valid submission for that task (0% valid submissions); *low (1–33%)* indicates a valid submission for between 1% and 33% of seeds; *medium (34–66%)* indicates a valid submission for between 34% and 66% of seeds; and *high (67–100%)* indicates a valid submission for more than 66% of seeds. Agents are sorted by the percentage of tasks in the *high* category first, with the combined *medium* and *high* percentage as a tiebreaker, highlighting those most reliable across the benchmark.

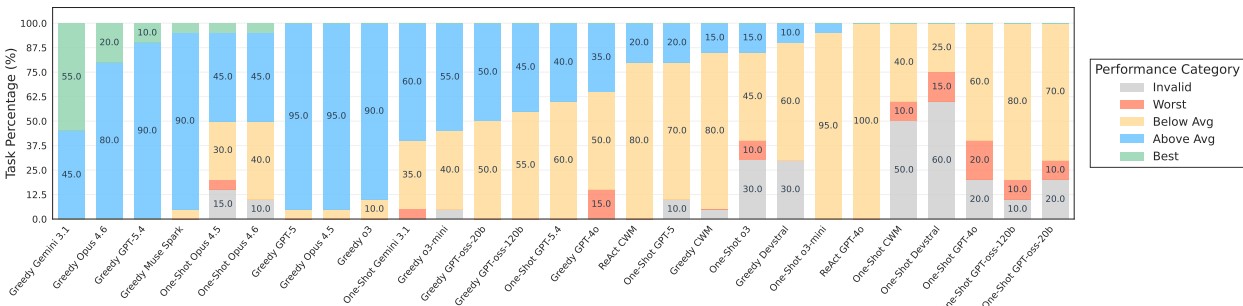

Figure 6: Performance distribution for the 27 agents evaluated. Each bar represents the percentage of tasks across all seeds for which a given agent falls into one of five performance categories: *invalid* (no valid submission for the task), *worst* (the lowest normalized score among all agents for the task), *below average* (normalized score below the mean but not the worst), *above average* (normalized score above the mean but not the best), and *best* (the highest normalized score for the task). Normalized scores are computed per task according to equations 2 and 3. Agents are sorted by the number of tasks for which they achieved the *best* performance first, with the combined count of *best* and *above average* performances as a tiebreaker, highlighting those with the most consistent top performance across the benchmark.

A breakdown of each agent's average normalized score per task is provided in Figure 6. For each agent, we report the percentage of tasks for which the agent yields one of five possible outcomes: (i) *invalid*, the agent does not submit a solution at all; (ii) *worst*, the agent produces the lowest score among all agents for that task; (iii) *below average*, the agents achieves a score below the average across all agents for that task; (iv) *above average*, the agent achieves a score above the average but is not the best; and (v) *best*, the agent achieves the highest score among all agents for that task. This breakdown highlights the distribution of each agent's performance across the benchmark tasks. Scores are normalized according to Equation 2, where $\phi_t$ is the march of 9s transform.

We report the mean valid submission rate defined in Equation 1 across the AIRS-BENCH tasks in Figure. 7. On average, only 66.4% of the total submissions are considered valid, suggesting that even submitting a valid solution stretches the capabilities of the agents. We also observe that reasoning models and the Greedy/ReAct scaffolds offer an advantage, as performance of these agents is superior.

In Figure 8 we report the average normalized scores according to Equations 2 and 3. A more detailed breakdown of the average scores per task is reported in Figure 9 with $\phi_t$ specified by Equation 3. The scores

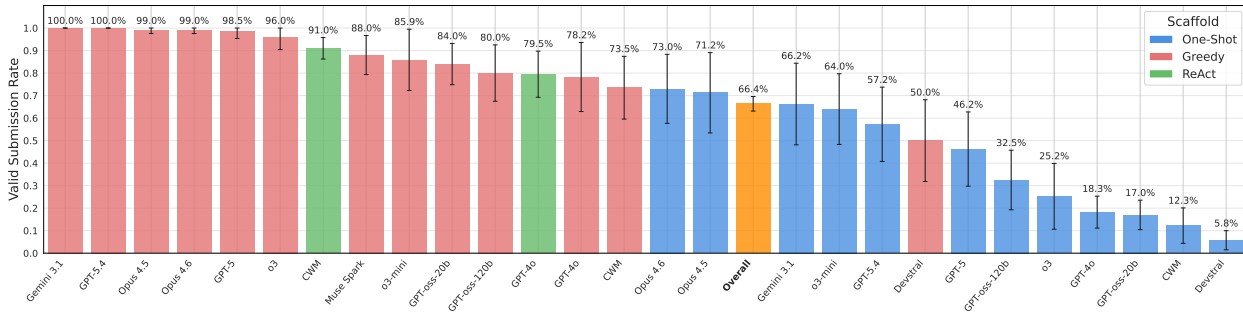

Figure 7: Mean valid submission rate (VSR) for the 27 agents evaluated, with error bars indicating the 95% confidence intervals. VSR is computed according to Eq. 1, with the overall average shown in orange.

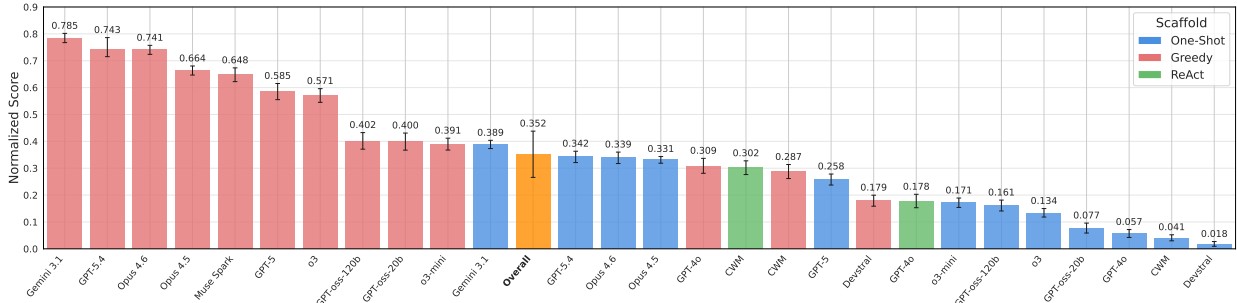

Figure 8: Average normalized scores for the 27 agents evaluated, with error bars indicating the 95% confidence intervals. Scores are computed according to Equations 2 and 3. The overall average normalized score across all runs and agents averages at 0.349, highlighting the challenging nature of AIRS-BENCH.

distribution in the figure reiterates the value of the AIRA-DOJO harness in supporting agents to develop better solutions, with Greedy scaffolds (in red) distributing closer to SOTA than One-Shot ones (in blue).

Figure 9 depicts normalized scores computed according to Equations 2 and 3: each row corresponds to a task and each point relates to the normalized score achieved by an agent on that task and averaged across multiple seeds. For each task we average the normalized score across all agents and seeds and we use this mean normalized score to sort tasks by difficulty level. We stack tasks from the easiest to the most difficult going from top to bottom. The mapping between task numbers appearing in the y axis of Figure 9 and task names can be found in Table 4 of Appendix C. Tasks with points to the right of the 1.0 line indicate that the average score of that agent on that task exceeds human SOTA, which is not necessarily the case for all tasks with at least one seed (i.e. agent run) exceeding human SOTA. Figure 12 in Appendix C similarly presents normalized scores across tasks and agents, but using in Equation 2 the identity transform from Equation 8 (in Appendix C) instead of the march of 9s transform from Equation 3.

Based on the task ranking from Figure 9, in Figure 10 we group the AIRS-BENCH tasks into four groups, each containing 5 tasks, and corresponding to an increasing level of difficulty: *easy* (tasks 1 to 5), *medium* (tasks 6 to 10), *hard* (tasks 11 to 15) and *expert* (tasks 16 to 20). Here we are averaging scores across 5 tasks, i.e. each point is the average over seeds of all 5 tasks in that difficulty bucket. We also observe in Figure 10 that while the normalized scores are all low and somewhat similar on the expert tasks, for the easier problems (and especially those in the easiest bucket), we see high variability between the scores that the agents achieved. The correspondence between task numbers and names is the same as the one in Figure 9 and can be found in Table 4 of Appendix C.

Finally, Elo ratings including human SOTA as an additional opponent alongside our agents are reported in Figure 11. The sizeable gap between the rating of the human SOTA player and the top-performing agent indicates that even the best agent is significantly below SOTA and the benchmark is very far from saturated.

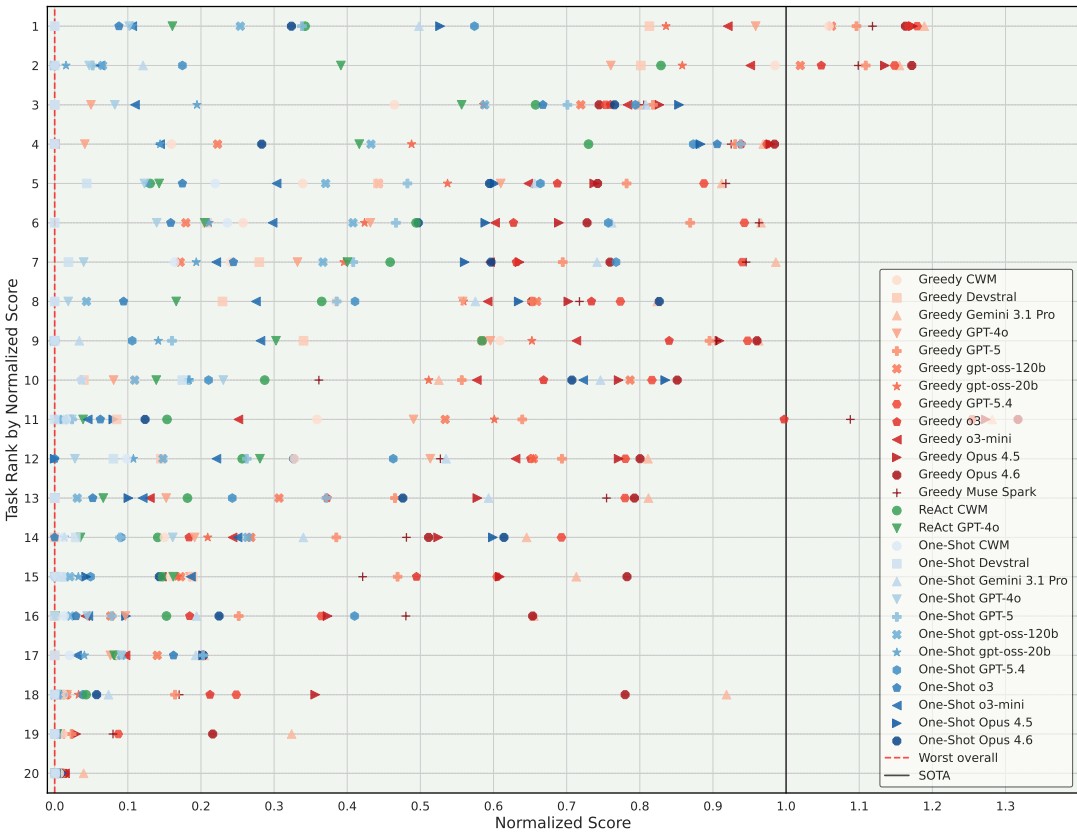

Figure 9: Average normalized scores with each row corresponding to an AIRS-Bench task and each point to an agent's normalized score for that task averaged across multiple seeds. For each task, the outcome of the worst-performing run is used as the baseline score. SOTA always corresponds to a normalized score of 1. Tasks are ranked from least to most difficult, from top to bottom, according to the average score across all agents. See Table 4 for the correspondence between tasks numbers on the y axis and names.

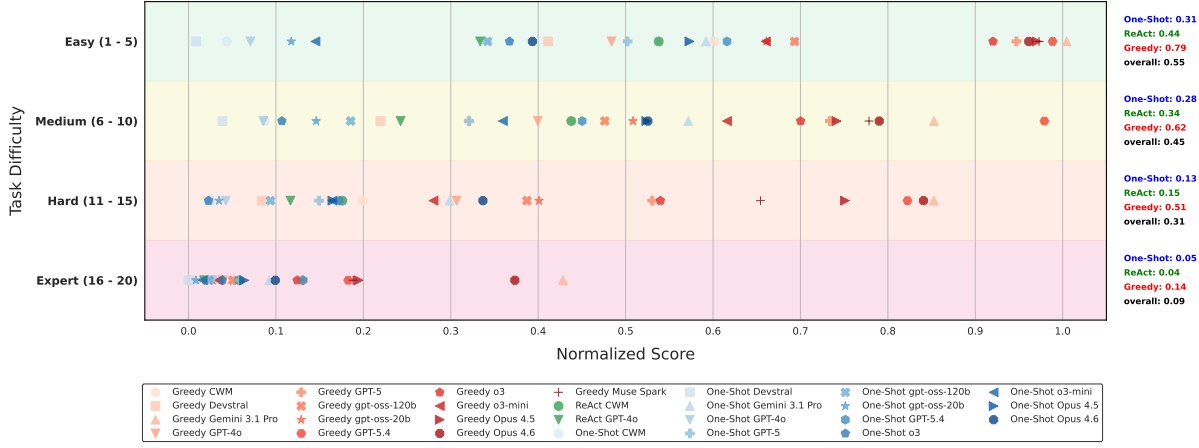

Figure 10: Normalized score per task difficulty level according to Equations 2-3. We divide the task ranking of Figure 9 into four categories with decreasing normalized scores: *easy*, *medium*, *hard* and *expert*.

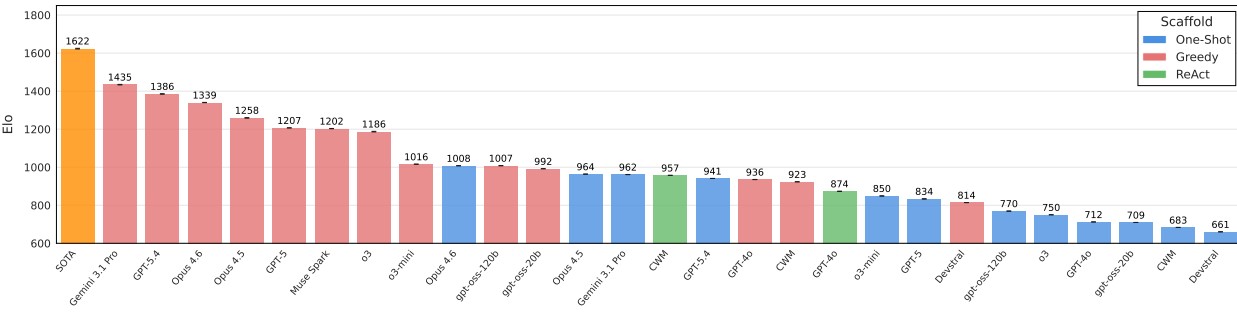

Figure 11: Elo ratings of all agents, estimated by fitting a Bradley–Terry model on the pairwise comparisons of agents' scores for each task. The human SOTA score is also included as an additional opponent. The Greedy scaffold outperforms other scaffolds in most cases. Bar height represents the median of a bootstrap distribution using 100 resamples, with the error bars representing the 95% confidence intervals.

## 6.2 Task Inspection: Success in Beating SOTA

Among the runs shown in Figures 9 and 12, we found cases where the agent's performance, at least in some of the seeds, was higher than the reported human SOTA, i.e. had a normalized score that was greater than 1. Overall, we identified 7 tasks where our agents surpass SOTA performance, as summarized in Table 3. We examined these cases in depth to better understand the solution produced by the agent, and how it manages to outperform the human SOTA. Below, we provide a breakdown for two notable case where an agent outperforms human SOTA with an original solution.

**Greedy Gemini 3.1 Pro on TimeSeriesForecastingRideshareMAE**.

This is a time-series forecasting task on the Monash rideshare dataset Godahewa et al. (2021), which contains hourly NYC ride-sharing demand series. The goal is to produce 48-step-ahead forecasts given historical observations, evaluated by mean absolute error (MAE).

The SOTA solution (Gong et al., 2025) integrates a prototype-guided dynamic normalization mechanism (ProtoNorm) into the Moirai foundation model (Woo et al., 2024), a transformer-based time-series foundation model pretrained on a large multi-dataset corpus that includes the rideshare dataset. ProtoNorm replaces standard LayerNorm with multiple dataset-aware normalization modules, selected dynamically via learned distributional prototypes. The resulting model, ProtoN-FM, is evaluated in an in-distribution setting (without fine-tuning) achieving an MAE of 1.185 on the rideshare dataset.

The Greedy Gemini 3.1 Pro agent takes a fundamentally different approach by constructing a task-specific forecasting pipeline trained on the rideshare data. The agent builds a DualForecaster architecture that combines two complementary forecasting backbones: DLinear Zeng et al. (2023), a lightweight linear decomposition model that separates trend and seasonal components, and PatchTST Nie et al. (2022), a patch-based transformer that segments the input series into subseries-level patches for efficient attention. Both branches are preceded by Reversible Instance Normalization (RevIN) (Kim et al., 2021) to handle distribution shift across time windows. The agent generates training examples via a sliding-window scheme with window size 192 and horizon 48, and employs 5-fold rolling cross-validation to ensure temporal integrity. This task-specific approach trades broad generalizability for stronger domain performance, achieving an MAE of 1.045, a 11.8% gain over the pretrained SOTA baseline.

**Greedy Gemini 3.1 pro on CoreferenceResolution-WinograndeAccuracy**.

This is a coreference resolution resolution task using the WinoGrande dataset Sakaguchi et al. (2021), which presents fill-in-the-blank sentences requiring commonsense reasoning to select the correct option from two candidates. The evaluation metric is accuracy on the validation split (test labels are typically hidden behind a leaderboard).

Table 3: AIRS-Bench tasks where at least one agent surpassed human SOTA in at least one run. All entries use the Greedy scaffold. Columns show: task name $t$; agent $a$; SOTA score $s_t^{\mathrm{sota}}$ and best agent score $s_t^a$; a brief comparison of the SOTA and agent solutions.

| Task | Agent | Score | Method |
|------|-------|-------|--------|
| CoreferenceResolution-WinograndeAccuracy | Gemini 3.1 Pro | **SOTA**: 0.854 
 **Agent**: 0.926 | **SOTA** (Lin et al., 2020): Fine-tune T5-3B in a text-to-text setup, scoring answer options by output token probabilities ("entailment" vs. "contradiction"). 

 **Agent**: Fine-tuned Qwen2.5-14B using LoRA fine-tuning. |
| GMolecularProperty-PredictionQm9MAE | Gemini 3.1 Pro | **SOTA**: 7.530 
 **Agent**: 7.157 | **SOTA** (Liao et al., 2023) Equivariant transformer leveraging efficient spherical convolutions and higher-degree geometric representations for accurate 3D molecular property prediction. 

 **Agent:** Hybrid physics-informed model combining a compositional ridge baseline with a DimeNet++ geometric GNN and a global thermodynamic MLP, trained on residuals to jointly capture atomic interactions and macroscopic scaling laws. |
| ReadingComprehension-SquadExactMatch | Gemini 3.1 Pro | **SOTA**: 0.858 
 **Agent**: 0.875 | **SOTA** (Zhu et al., 2024): ALBERT-based QA model using mixed-precision training and context-splitting 

 **Agent:** DeBERTa-based extractive QA model with learnable last-layer fusion, multi-sample dropout ensembling for robust span prediction, and label-smoothed training with vectorized span decoding for efficient inference. |
| TextualClassificationSick-Accuracy | Opus 4.6 | **SOTA**: 0.905 
 **Agent**: 0.937 | **SOTA** (Kalouli et al., 2023): Vanilla fine-tuning of RoBERTa. 

 **Agent**: Off-the-shelf NLI-tuned DeBERTa-v3-large and RoBERTa-large ensemble |
| TextualSimilaritySick-SpearmanCorrelation | Gemini 3.1 Pro | **SOTA**: 0.854 
 **Agent**: 0.909 | **SOTA** (Huang et al., 2024a): Finetuned RoBERTa-large and novel loss function (CoSENT). 

 **Agent**: DeBERTa-v3-large cross-encoder with attention pooling, ColBERT-style token alignment, pairwise ranking loss, R-Drop, and entropy-weighted test-time augmentation. |
| TimeSeriesForecasting-RideshareMAE | Gemini 3.1 Pro | **SOTA**: 1.185 
 **Agent**: 1.045 | **SOTA** (Gong et al., 2025): General transformer-based time-series foundation model (pretrained on multi-dataset corpus, not specifically finetuned). 

 **Agent**: Dual-branch DLinear + PatchTST hybrid with ReVIN normalization, trained on historical sliding windows. |
| U0MolecularProperty-PredictionQm9MAE | Muse Spark | **SOTA**: 5.850 
 **Agent**: 5.722 | **SOTA (Schütt et al., 2021):** PaiNN equivariant message passing GNN. 

 **Agent:** Hybrid physics-informed model combining a compositional ridge baseline with a DimeNet++ geometric GNN and gradient boosting on Coulomb/Bag-of-Bonds features. |

The SOTA solution Lin et al. (2020) fine-tunes T5-3B Raffel et al. (2020) in a text-to-text setup, decomposing each problem into two hypothesis–premise pairs and scoring them via softmax over the "entailment" and "contradiction" target tokens. The approach achieves 85.4% accuracy on the validation set.

The Greedy Gemini 3.1 Pro agent adopts a similar paradigm by fine-tuning a pretrained language model on the WinoGrande training set, but applies it at greater scale with modern parameter-efficient techniques. The agent selects Qwen2.5-14B Qwen Team (2025) as the backbone and applies LoRA adaptation Hu et al. (2022) with rank 64 across all attention and MLP projections, enabling efficient fine-tuning of the 14B-parameter model within the compute budget. To mitigate positional bias inherent in multiple-choice tasks, the agent augments the training data by duplicating each example with swapped option order, ensuring the model learns to reason about content rather than option position. At inference, the agent applies test-time augmentation by averaging predictions from both the original and swapped option orderings, further reducing positional artifacts. This pipeline achieves a validation accuracy of 92.6%, representing an 8.4% relative improvement over the SOTA.

## 7 Conclusion

We introduced AIRS-Bench, a benchmark designed to rigorously evaluate the autonomous research capabilities of LLM agents in machine learning. Our benchmark covers 20 diverse, non-contaminated tasks spanning multiple domains, and is specifically constructed to assess agents across the full research workflow—from ideation and methodology design to experimentation and iterative refinement—without access to baseline code. Our results indicate a high variability in task performance, depending on both the LLM that the agent uses, and the scaffold it is based on. For most tasks, even the best performing agent is still significantly behind the human SOTA, showing that the benchmark is far from saturated. For a few tasks, our top agents managed to identify a solution outperforming the human SOTA. It would be interesting to see how much AI Research Agents can push the state-of-the-art further.[6]

We gathered a number of useful takeaways while building AIRS-Bench and evaluating a number of agents across its tasks:

- Gaps in community infrastructure: within the current state of AI research, the task of tracking up-to-date SOTA became more challenging than ever. Both the growing amounts of paper submissions,[7] the high compute cost of reproducing experiments on large models and the lack of unified platforms to represent results contribute to the situation. A new shared space with standardized format, updates, and machine-readable configurations for all published machine learning research is needed.

- Performance gaps: several main factors cause the performance of the agents to be lower than it could be. The combinations of base LLMs with different scaffolds can lead to 1) problems with formatting, specifically submitting the correct solution after the experiments, 2) problems with saving intermediate results, 3) performance deterioration on main capabilities due to the context overflow. Longer agentic traces also lead to increased probabilities of misaligned behaviours and accumulated issues around code edits and debugging, which make it difficult to adhere to the methodology of the ML experimentation.

- Human bottlenecks: for the work to be scaled across new domains and a bigger volume of tasks, automatic task onboarding pipelines will be needed. Current human validation procedures prevent the expansion at scale.

- Role of restrictions: The ML benchmarks for AI Agents tend to be computationally costly, both on the training and inference side. Given the significant resource constraints faced in agentic evaluations—such as computational costs, time limits, and token usage—we acknowledge the possible role of restrictions (e.g. 24h time limit) in the obtained results. Benchmark methodology commonly faces the choice of either evaluating the systems in the very well-defined restricted conditions or lifting most of them and comparing the best obtained results. Although we adhere to the first choice for the sake of future extensive ablations, lifting certain restrictions could enable more flexible and efficient agent behaviors in the future.

Our results demonstrate that scaffold design significantly impacts agent solution quality, highlighting opportunities to improve performance through algorithms that better leverage test-time compute. We release AIRS-Bench to help identify performance gaps in AI Research Agents and catalyze the development of better methods for accelerating scientific progress. As agentic capabilities advance, continued benchmark development will be essential. We hope this benchmark fosters transparency, reproducibility, and rigorous, standardized evaluation of LLM agents in frontier research contexts.

---

[6]Note that even in the cases where the agent outperforms human SOTA, it is still interesting to see how far ahead the agent's solution is over the human baseline. Hence, tasks where the agent normalized score exceeds 1.0 are still interesting benchmark tasks.

[7]https://forum.cspaper.org/topic/76/submission-tsunami-at-neurips-2025-is-peer-review-about-to-collapse/2

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

## A  Appendix

## B  Task Selection

We constructed AIRS-BENCH by downsampling a pool $\mathcal{F}$ of approximately 100 tasks to a representative subset $\mathcal{S}$ of 20 tasks. The reduction to 20 tasks was implemented to substantially decrease GPU requirements and enable faster benchmarking. The AIRS-BENCH subset was selected to closely mirror the full pool $\mathcal{F}$ according to three key criteria:

- **Agent performance**: each agent's average score on AIRS-BENCH is as close as possible to their average score on the full benchmark.

- **Category distribution**: the proportion of tasks from each of the 7 categories in AIRS-BENCH is as close as possible to that of the full pool.

- **Relative ranking fidelity**: the ranking of agents by performance is preserved between AIRS-BENCH and the full pool.

For each agent $a$, we compute the mean normalized score $\overline{\mathrm{NS}}^a_{\mathcal{F}}$ over $\mathcal{F}$ and $\overline{\mathrm{NS}}^a_{\mathcal{S}}$ over AIRS-BENCH, where

$$\overline{\mathrm{NS}}^a_{\mathcal{F}} = \frac{1}{|\mathcal{F}|} \sum_{t \in \mathcal{F}} \mathrm{NS}^a_t \qquad \overline{\mathrm{NS}}^a = \frac{1}{|\mathcal{S}|} \sum_{t \in \mathcal{S}} \mathrm{NS}^a_t \tag{6}$$

where the normalized score $\mathrm{NS}^a_t$ is defined in Equation 2 (see Section 5.2) and is the performance score of agent $a$ on task $t$. The AIRS-BENCH subset is selected to minimize the mean absolute error (MAE) between $\overline{\mathrm{NS}}^a_{\mathcal{F}}$ and $\overline{\mathrm{NS}}^a_{\mathcal{S}}$ across all agents:

$$\mathrm{MAE} = \frac{1}{|A|} \sum_{a \in A} \left| \overline{\mathrm{NS}}^a_{\mathcal{F}} - \overline{\mathrm{NS}}^a_{\mathcal{S}} \right| \tag{7}$$

where $A$ is the set of all agents. To ensure representative coverage, $\mathcal{F}$ is partitioned into four difficulty bands (*easy*, *medium*, *hard*, *expert*), based on their relative ranking by average normalised score (see Section 6.1) and each containing approximately 25 tasks. AIRS-BENCH is constructed by sampling a fixed number of tasks from each band. Four candidate difficulty band distributions were evaluated:

- **Uniform**: 5 tasks each from *easy*, *medium*, *hard*, and *expert* bands ($\{5, 5, 5, 5\}$).

- **Medium-skewed**: 4 *easy*, 7 *medium*, 5 *hard*, and 4 *expert* tasks ($\{4, 7, 5, 4\}$).

- **Center-skewed**: 4 *easy*, 6 *medium*, 6 *hard*, and 4 *expert* tasks ($\{4, 6, 6, 4\}$).

- **Medium-heavy**: 3 *easy*, 8 *medium*, 6 *hard*, and 3 *expert* tasks ($\{3, 8, 6, 3\}$).

The final band allocation was selected by choosing the configuration that minimized the mean absolute error (MAE) between agent scores on the subset and the full benchmark. The search for the optimal subset was performed using three subset selection algorithms:

- **Random search**: samples ten thousand candidate subsets, each respecting the band constraints, and retains the subset with the lowest MAE.

- **Simulated annealing**: iteratively swaps tasks within bands, accepting both improvements and, with decreasing probability, worse solutions to escape local minima.

- **Genetic algorithm**: evolves a population of candidate subsets through tournament selection, single-point crossover ($p = 0.7$), and mutation ($p = 0.2$) minimizing MAE over generations.

Across all 12 possible ({algorithm} $\times$ {band distribution}) combinations, the best-performing configuration was obtained using the genetic algorithm with a medium-skewed band allocation, achieving a minimum MAE of $4.0 \times 10^{-3}$. Other competitive configurations included both uniform and skewed band distributions, with MAE values ranging from $4.6 \times 10^{-3}$ to $7.9 \times 10^{-3}$.

To validate the fidelity of AIRS-BENCH, we compared agent mean normalized scores and their 95% confidence intervals on AIRS-BENCH versus the full pool $\mathcal{F}$. The results demonstrate that agent rankings and score gaps are faithfully preserved, with overall mean scores and confidence intervals nearly identical between the two sets: the difference in average score between the subset and the full benchmark never exceeds 0.02 in absolute value. This confirms that the selection criterion and stratified sampling approach yield a lightweight benchmark that maintains the discriminative power and ranking structure of the original pool, while allowing efficient evaluation.

## C  Additional Results

We consider an additional normalized score (see Eq. 2) which employs the identity transform

$$\phi_t(s) = \mathcal{I}(s) = s \tag{8}$$

In contrast to Eq. 3, this approach directly uses raw scores. With this transform, the normalized score $\text{NS}_t^a$ linearly reflects the agent's progress between the worst observed solution and the human SOTA for each task. This is simple and interpretable, but may not always reflect meaningful progress when the metric is highly non-linear or when the gap to the optimal score is very small. For completeness, we show the average normalized score using this transform in Figure 12, and provide a breakdown of the scores by difficulty level in Figure 13.

For better readability, Table 4 provides the mapping between task numbers shown on the y-axis of Figures 9-12 and their corresponding task names, as well as their average score across all seeds and agents.

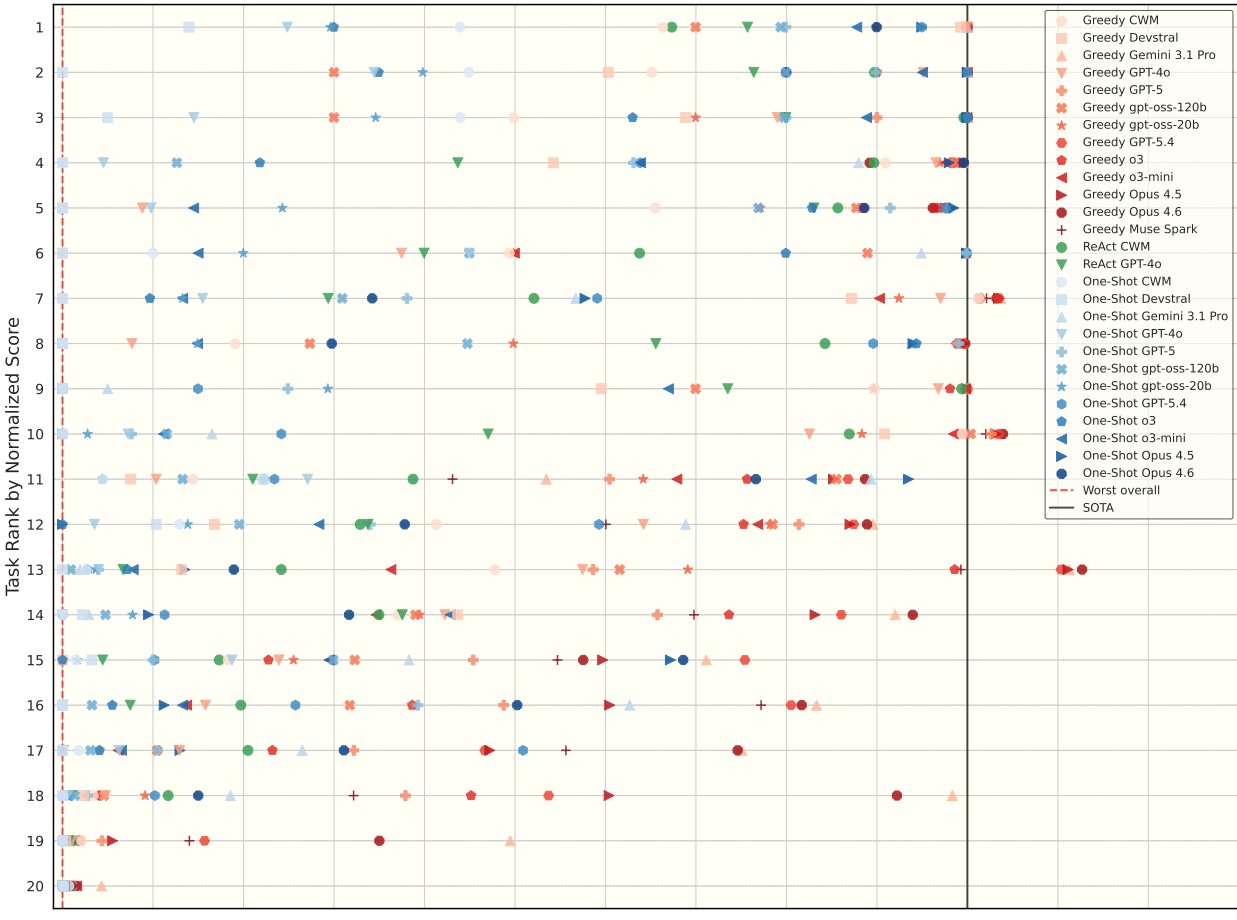

Figure 12: Normalized score per task averaged over seeds, computed according to Equations 2- 8. For each task, the outcome of the worst-performing run is used as the baseline score $s_t^{\min}$. SOTA always corresponds to a normalized score of 1. Tasks are ranked in decreasing order according to the average score across all agents. See Table 4 for the correspondence between tasks ranking and name.

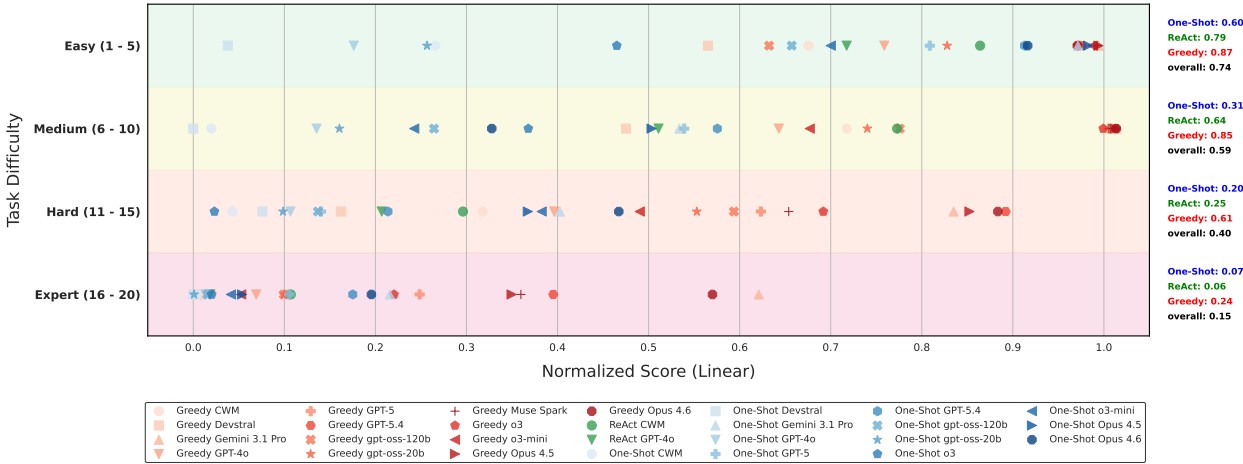

Figure 13: Normalized score per task difficulty level according to Equations 2-8. We divide the task ranking of Figure 12 into four categories with decreasing normalized scores: *easy, medium, hard* and *expert*.

Table 4: Side-by-side ranking of tasks by average normalized score with **march of 9's** and **identity** transforms, as reported in Figure 9, and Figure 12, respectively. Color represents progressively harder tasks, from *easy* to *expert*.

| Rank | Task (March of 9's) | $NS_t^a$ | Task (Identity) | $NS_t^a$ |
|---|---|---|---|---|
| 1 | TextualSimilaritySickSpearmanCorrelation | 0.64 | R2AbsMolecularPropertyPredictionQm9MAE | 0.79 |
| 2 | TextualClassificationSickAccuracy | 0.56 | U0MolecularPropertyPredictionQm9MAE | 0.77 |
| 3 | TimeSeriesForecastingSolarWeeklyMAE | 0.55 | GMolecularPropertyPredictionQm9MAE | 0.77 |
| 4 | TimeSeriesForecastingRideshareMAE | 0.53 | GraphRegressionZincMae | 0.69 |
| 5 | R2AbsMolecularPropertyPredictionQm9MAE | 0.49 | TimeSeriesForecastingSolarWeeklyMAE | 0.69 |
| 6 | U0MolecularPropertyPredictionQm9MAE | 0.49 | TimeSeriesForecastingKaggleWebTrafficMASE | 0.66 |
| 7 | GMolecularPropertyPredictionQm9MAE | 0.47 | TextualSimilaritySickSpearmanCorrelation | 0.63 |
| 8 | GraphRegressionZincMae | 0.44 | TimeSeriesForecastingRideshareMAE | 0.56 |
| 9 | CvMolecularPropertyPredictionQm9MAE | 0.43 | CvMolecularPropertyPredictionQm9MAE | 0.56 |
| 10 | ReadingComprehensionSquadExactMatch | 0.42 | TextualClassificationSickAccuracy | 0.55 |
| 11 | CoreferenceResolutionWinograndeAccuracy | 0.40 | ReadingComprehensionSquadExactMatch | 0.48 |
| 12 | SentimentAnalysisYelpReviewFullAccuracy | 0.40 | SentimentAnalysisYelpReviewFullAccuracy | 0.47 |
| 13 | QuestionAnsweringEli5Rouge1 | 0.29 | CoreferenceResolutionWinograndeAccuracy | 0.40 |
| 14 | QuestionAnsweringDuoRCAccuracy | 0.26 | CoreferenceResolutionSuperGLUEWSCAccuracy | 0.37 |
| 15 | CoreferenceResolutionSuperGLUEWSCAccuracy | 0.22 | QuestionAnsweringDuoRCAccuracy | 0.30 |
| 16 | CodeRetrievalCodeXGlueMRR | 0.17 | QuestionAnsweringEli5Rouge1 | 0.30 |
| 17 | TimeSeriesForecastingKaggleWebTrafficMASE | 0.13 | CodeRetrievalCodeXGlueMRR | 0.21 |
| 18 | MathQuestionAnsweringSVAMPAccuracy | 0.12 | MathQuestionAnsweringSVAMPAccuracy | 0.19 |
| 19 | QuestionAnsweringFinqaAccuracy | 0.03 | QuestionAnsweringFinqaAccuracy | 0.05 |
| 20 | CodeGenerationAPPSPassAt5 | 0.01 | CodeGenerationAPPSPassAt5 | 0.01 |

## C.1 Task run files

### C.1.1 project_description.md

## Overview

### Task Description

This is a Machine Learning project and your goal is to build a model that solves the project's TASK following the instructions provided below.

> TASK: Your task is to solve math word problems. Each example presents a short story followed by a specific question. Your task is to read the text and predict the correct numerical answer. Your predictions will be scored against the Answer column of the test set.

## Data

### Dataset Structure

The default config of SVAMP has the following structure. Here is a description of the contents of each column including their name, what they contain, and the data type:

```
{
  "ID": string,                 # Unique identifier for the problem
  "Body": string,               # The narrative or context text describing the scenario
  "Question": string,           # The actual math question asked based on the Body
  "Equation": string,           # The ground-truth equation used to compute the answer
  "Answer": string,             # The correct numerical solution to the problem (as text)
  "Type": string,               # The problem category/type. Is one of ["Subtraction", "Addition",
                                # "Common-Division", "Multiplication"].
  "question_concat": string     # Concatenation of Body and Question into one text field
}
```

An example entry in the train dataset contains the following:

- ID: "chal-777"

- Body: "There are 87 oranges and 290 bananas in Philip's collection. If the bananas are organized into 2 groups and oranges are organized into 93 groups"

- Question: "How big is each group of bananas?"

- Equation: "( 290.0 / 2.0 )"

- Answer: "145"

- Type: "Common-Division"

- question_concat: "There are 87 oranges and 290 bananas in Philip's collection. If the bananas are organized into 2 groups and oranges are organized into 93 groups How big is each group of bananas?"

**Note:** The scoring columns (Equation, Answer, Type) have been removed from the test data. Ensure you preserve the original row order of the test set when producing predictions.

Train data can be loaded using:

```
from datasets import load_from_disk
train_dataset = load_from_disk('./data/train')
```

Test data can be loaded using:

```
from datasets import load_from_disk
test_dataset = load_from_disk('./data/test')
```

## Submission File

The submission file should contain the selected answers for the test set. The submission file should be a .csv file named `submission.csv` with the following header:

```
Answer
```

And it should be of shape `(300,1)` — one prediction per row, aligned with the test set order. Each row must contain the numerical answer to the question. Do not include any extra columns or indices.

## Evaluation

The evaluation will be performed on the `submission.csv` file you have submitted using the Accuracy metric. Evaluation will be done using the following script:    30

```python
1   #!/usr/bin/env python3
2   import argparse, json
3   import numpy as np
4   import pandas as pd
5   from datasets import load_from_disk
6
7
8   def load_test_answers(path='./data/test_with_labels'):
9       """
10      Load answers for evaluation (string labels).
11      """
12      ds = load_from_disk(path)
13      if "Answer" not in ds.features:
14          raise RuntimeError(
15              "Test data does not contain 'Answer'. "
16              "Use a labeled copy like './data/test_with_labels'."
17          )
18      answers = list(ds["Answer"])
19      return answers
20
21
22  def evaluate(predictions, labels):
23      """
24      Returns a dict of metric_name -> value
25      """
26      correct = np.fromiter(
27          (int(p) == int(y) for p, y in zip(predictions, labels)), dtype=bool
28      )
29      return {"Accuracy": float(correct.mean())}
30
31
32  def _cli():
33      p = argparse.ArgumentParser(
34          description="Evaluate predictions for SVAMP test split using Accuracy."
35      )
36      p.add_argument("--submission-file", default="submission.csv",
37                      help="Path to CSV file containing predictions.")
38      a = p.parse_args()
39
40      print("Loading test set labels...")
41      labels = load_test_answers()
42      n_test_samples = len(labels)
43      print(f"Loaded {n_test_samples} labels.")
44
45      print(f"Loading predictions from: {a.submission_file}")
46      try:
47          # Assuming CSV has no header and contains only prediction values
48          # Adjust if your submission format is different (e.g., has headers, specific columns)
49          submission_df = pd.read_csv(a.submission_file, header=0)
50          preds = submission_df.values.squeeze()
51          if preds.shape[0] != n_test_samples:
52              raise ValueError(
53                  f"Submission file row count ({preds.shape[0]}) "
54                  f"does not match test set size ({n_test_samples})."
55              )
56      except FileNotFoundError:
57          p.error(f"Submission file not found: {a.submission_file}")
58      except Exception as e:
59          p.error(f"Error loading submission_file: {e}")
60
61      print("Evaluating predictions...")
```

```
62      result = evaluate(preds, labels)
63
64      print("\n--- EVALUATION RESULT ---")
65      print(json.dumps(result, indent=2))
66
67
68  if __name__ == '__main__':
69      _cli()
```

### C.1.2 evaluate.py

```python
1   #!/usr/bin/env python3
2   import argparse, json
3   import numpy as np
4   import pandas as pd
5   from datasets import load_from_disk
6
7
8   def load_test_answers(path='./data/test_with_labels'):
9       """
10      Load answers for evaluation (string labels).
11      """
12      ds = load_from_disk(path)
13      if "Answer" not in ds.features:
14          raise RuntimeError(
15              "Test data does not contain 'Answer'. "
16              "Use a labeled copy like './data/test_with_labels'."
17          )
18      answers = list(ds["Answer"])
19      return answers
20
21
22  def evaluate(predictions, labels):
23      """
24      Returns a dict of metric_name -> value
25      """
26      correct = np.fromiter(
27          (int(p) == int(y) for p, y in zip(predictions, labels)), dtype=bool
28      )
29      return {"Accuracy": float(correct.mean())}
30
31
32  def _cli():
33      p = argparse.ArgumentParser(
34          description="Evaluate predictions for SVAMP test split using Accuracy."
35      )
36      p.add_argument("--submission-file", default="submission.csv",
37                     help="Path to CSV file containing predictions.")
38      a = p.parse_args()
39
40      print("Loading test set labels...")
41      labels = load_test_answers()
42      n_test_samples = len(labels)
43      print(f"Loaded {n_test_samples} labels.")
44
45      print(f"Loading predictions from: {a.submission_file}")
46      try:
47          # Assuming CSV has no header and contains only prediction values
48          # Adjust if your submission format is different (e.g., has headers, specific columns)
49          submission_df = pd.read_csv(a.submission_file, header=0)
50          preds = submission_df.values.squeeze()
```

```python
51            if preds.shape[0] != n_test_samples:
52                raise ValueError(
53                    f"Submission file row count ({preds.shape[0]}) "
54                    f"does not match test set size ({n_test_samples})."
55                )
56        except FileNotFoundError:
57            p.error(f"Submission file not found: {a.submission_file}")
58        except Exception as e:
59            p.error(f"Error loading submission_file: {e}")
60
61        print("Evaluating predictions...")
62        result = evaluate(preds, labels)
63
64        print("\n--- EVALUATION RESULT ---")
65        print(json.dumps(result, indent=2))
66
67
68  if __name__ == '__main__':
69      _cli()
```

### C.1.3 metadata.yaml

```yaml
metric_lower_is_better: false
file_export_globs:
  - submission.csv
container_python_requirements:
  - datasets==4.0.0
evaluate_container_python_requirements:
  - datasets==4.0.0
logging_info:
  name: MathQuestionAnsweringSVAMPAccuracy
  category: Math
  research_problem: Math Question Answering
  output_type: text-generation
  dataset: ChilleD/SVAMP
  metric: Accuracy
  input_columns:
    - question_concat
  scoring_column: Answer
  shape: 300,1
  config: default
  train_split: train
  test_split: test
  custom_gold_labels: false
  custom_rad_class: false
  sota:
    - sota_paper_title: 'Achieving >97% on GSM8K: Deeply Understanding the Problems
        Makes LLMs Better Solvers for Math Word Problems'
      sota_paper_url: https://arxiv.org/pdf/2404.14963v5
      sota_score: 0.942
      sota_notes: DUP is a prompting template. Result provided is for GPT-4 with the
        GUP prompting template.
      sota_year: 2026
      sota_venue: Frontiers of Computer Science
  dataset_paper_url: https://arxiv.org/abs/2103.07191
  estimated_worst_score: 0.0
  optimal_score: 1.0
```

# D   Harness Setup

Table 5: Resources and constraints comparison between AIRA-DOJO and MLGYM.

| | AIRA-dojo | MLGym |
|---|---|---|
| **Time/number of steps limit** | 24 hours (up to ∼36 including evaluation time) | 1M steps / 24 hours |
| **Can exit early** | No | No |
| **evaluate.py file is visible** | Yes | Yes |
| **Test set with labels is visible** | No | No |
| **Validation script** | Agent codes during run | Agent codes during run |
| **Validation splits** | Cross-validation (Greedy only) | Classical split 70-30 |
| **Scaffold implemented** | Greedy (AIDE) | ReAct |
| **Last submission valid always?** | No | No |
| **Pretrained models access** | Yes | Yes |
| **Num of steps / nodes captured** | Yes | Yes |
| **All validation scores captured** | Yes | Yes |
| **Every solution scored on test set** | Yes | Only final submitted solution |
| **Time limit per solution** | 4 hours | 1 hour |
| **Dummy submission provided** | No | No |
| **Num GPUs** | 1 H200 / run | 1 H200 / run |
| **Internet access** | Yes (HF_OFFLINE=True, but agent can set it to False) | Yes (can be turned off) |
| **How is evaluate.py provided** | In prompt | In shared workspace |
| **Python version** | 3.10 | 3.10 |
| **Datasets library version** | 3.5.1 (upgrade to 4.0.0) | 4.0.0 |

## D.1   MLGym system prompt

```
SETTING: You are an autonomous Machine Learning Researcher, and you're working directly in the
    command line with a special interface.

The special interface consists of a file editor that shows you 1000 lines of a file at a time.
In addition to typical bash commands, you can also use the following commands to help you navigate
    and edit files.

COMMANDS:
open:
  docstring: opens the file at the given path in the editor. If line_number is provided, the window
    will be move to include that line
  signature: open "<path>" [<line_number>]
  arguments:
    - path (string) [required]: the path to the file to open
    - line_number (integer) [optional]: the line number to move the window to (if not provided, the
    window will start at the top of the file)

goto:
  docstring: moves the window to show <line_number>
  signature: goto <line_number>
  arguments:
    - line_number (integer) [required]: the line number to move the window to
```

```
scroll_down:
  docstring: moves the window down 1000 lines
  signature: scroll_down

scroll_up:
  docstring: moves the window down 1000 lines
  signature: scroll_up

create:
  docstring: creates and opens a new file with the given name
  signature: create <filename>
  arguments:
    - filename (string) [required]: the name of the file to create

search_dir:
  docstring: searches for search_term in all files in dir. If dir is not provided, searches in the
    current directory
  signature: search_dir <search_term> [<dir>]
  arguments:
    - search_term (string) [required]: the term to search for
    - dir (string) [optional]: the directory to search in (if not provided, searches in the current
    directory)

search_file:
  docstring: searches for search_term in file. If file is not provided, searches in the current
    open file
  signature: search_file <search_term> [<file>]
  arguments:
    - search_term (string) [required]: the term to search for
    - file (string) [optional]: the file to search in (if not provided, searches in the current
    open file)

find_file:
  docstring: finds all files with the given name in dir. If dir is not provided, searches in the
    current directory
  signature: find_file <file_name> [<dir>]
  arguments:
    - file_name (string) [required]: the name of the file to search for
    - dir (string) [optional]: the directory to search in (if not provided, searches in the current
    directory)

edit:
  docstring: replaces lines <start_line> through <end_line> (inclusive) with the given text in the
    open file. The replacement text is terminated by a line with only end_of_edit on it. All of the
    <replacement text> will be entered, so make sure your indentation is formatted properly. Python
    files will be checked for syntax errors after the edit. If the system detects a syntax error,
    the edit will not be executed. Simply try to edit the file again, but make sure to read the
    error message and modify the edit command you issue accordingly. Issuing the same command a
    second time will just lead to the same error message again.
  signature: edit <start_line>:<end_line>
<replacement_text>
end_of_edit
  arguments:
    - start_line (integer) [required]: the line number to start the edit at
    - end_line (integer) [required]: the line number to end the edit at (inclusive)
    - replacement_text (string) [required]: the text to replace the current selection with
```

```
insert:
  docstring: inserts the given text after the specified line number in the open file. The text to
      insert is terminated by a line with only end_of_insert on it. All of the <text_to_add> will be
      entered, so make sure your indentation is formatted properly. Python files will be checked for
      syntax errors after the insertion. If the system detects a syntax error, the insertion will not
      be executed. Simply try to insert again, but make sure to read the error message and modify the
      insert command you issue accordingly.
  signature: insert <line_number>
<text_to_add>
end_of_insert
  arguments:
    - line_number (integer) [required]: the line number after which to insert the text
    - text_to_add (string) [required]: the text to insert after the specified line

submit:
  docstring: submits your current code for evaluation on the test set and allows you to continue
      hill climbing. The test score will be hidden to prevent overfitting, but you'll get
      confirmation if the submission was valid. Only submit within triple quotes (```) should be
      executed - nothing else. Otherwise it will lead to parsing errors
  signature: submit
```

Please note that THE EDIT and INSERT COMMANDS REQUIRES PROPER INDENTATION.
If you'd like to add the line '          print(x)' you must fully write that out, with all those
    spaces before the code! Indentation is important and code that is not indented correctly will
    fail and require fixing before it can be run.

RESPONSE FORMAT:
Your shell prompt is formatted as follows:
(Open file: <path>) <cwd> $

You MUST format your output using EXACTLY two fields: DISCUSSION and command.
Your output should ALWAYS include *one* DISCUSSION section and *one* command section in EXACTLY
    this format:

DISCUSSION
[Your reasoning, thoughts, and plan for this step. Be specific about what you're trying to
    accomplish.]

```

[single command to execute]
```

CRITICAL FORMATTING RULES:
1. Start with "DISCUSSION" (no quotes, no extra formatting)
2. Write your thoughts and reasoning
3. Add a single line with three backticks: ```
4. Write exactly ONE command
5. End with three backticks: ```
6. NO text after the closing backticks
7. NO multiple commands in one response
8. NO multiple code blocks

EXAMPLE OF CORRECT FORMAT:
DISCUSSION

I need to explore the current directory to understand the project structure and identify the data files.

```
ls -la
```

You should only include a *SINGLE* command in the command section and then wait for a response from the shell before continuing with more discussion and commands. Everything you include in the DISCUSSION section will be saved for future reference. Please do not include any DISCUSSION after your action.
If you'd like to issue two commands at once, PLEASE DO NOT DO THAT! Please instead first submit just the first command, and then after receiving a response you'll be able to issue the second command.
You're free to use any other bash commands you want (e.g. find, grep, cat, ls, cd) in addition to the special commands listed above.
However, the environment does NOT support interactive session commands (e.g. python, vim), so please do not invoke them.

MACHINE LEARNING WORKFLOW:
Your goal is to achieve the best possible score on a hidden test set. Follow this systematic approach:

1. EXPLORATION PHASE:
   - Understand the task by examining data files, baseline scripts, and evaluation metrics
   - Identify the problem type (classification, regression, etc.)
   - Analyze data structure, features, and target variables

2. VALIDATION SETUP:
   - Create a proper train/validation split from the training data
   - If a separate validation set already exists, use it instead of creating your own
   - Set up evaluation code to measure performance on your validation set
   - Your validation code should be separate from evaluate.py provided but it can borrow motivation
     from how you'll be evaluated with evaluate.py when you do your final submission

3. BASELINE IMPLEMENTATION:
   - Start with the provided baseline script if available
   - Understand the baseline approach and its performance
   - Ensure you can reproduce baseline results

4. ITERATIVE IMPROVEMENT (MOST IMPORTANT PHASE):
   - Make incremental improvements to your model/approach
   - After each change, retrain your model and evaluate on validation set
   - Keep track of what works and what doesn't
   - Try different approaches: feature engineering, model architectures, hyperparameters
   - CONTINUE EXPERIMENTING even after finding a working solution - this is just the beginning!
   - Use hillclimbing: always try to improve your current best solution
   - Track your best validation score and keep trying to beat it

5. SUBMISSION AND CONTINUED HILL CLIMBING:
   - The `submit` command now allows you to test your solution without ending the session!
   - When you submit, your code is evaluated on the hidden test set, but the actual score is hidden
     to prevent overfitting
   - You'll receive confirmation that your submission was valid, then you can continue improving
   - **STRATEGY**: Submit whenever you have a working solution, then keep hill climbing to improve
     it further
   - Use multiple submissions throughout your session to test different approaches

- **CONTINUE AFTER SUBMIT**: After each submit, keep experimenting with new architectures, features, and techniques
- The goal is to maximize your compute time for hill climbing rather than just finding one working solution

IMPORTANT TIPS:
1. Always start by trying to understand the baseline script if available. This will give you an idea of one possible solution for the task and the baseline scores that you have to beat.

2. If you run a command and it doesn't work, try running a different command. A command that did not work once will not work the second time unless you modify it!

3. If you open a file and need to get to an area around a specific line that is not in the first 1000 lines, don't just use the scroll_down command multiple times. Instead, use the goto <line_number> command. It's much quicker.

4. Always make sure to look at the currently open file and the current working directory (which appears right after the currently open file). The currently open file might be in a different directory than the working directory! Note that some commands, such as 'create', open files, so they might change the current  open file.

5. When editing files, it is easy to accidentally specify a wrong line number or to write code with incorrect indentation. Always check the code after you issue an edit to make sure that it reflects what you wanted to accomplish. If it didn't, issue another command to fix it.

6. You have a limited number of actions/steps you can take in the environment. The current step and remaining number of steps will be given after every action. Use the remaining steps wisely by continuously hill climbing and making multiple submissions to test your improvements!

7. Your each action should take less than 3600 seconds to complete. If your action doesn't finish within the time limit, it will be interrupted.

8. VALIDATION IS KEY: Always use your own validation set (create or use if one exists in data directory) to evaluate model performance. Use the `submit` command to test your submission.csv with the hidden test set whenever you have a working solution, then continue improving. Note that test scores will be hidden to prevent overfitting, but your submissions will be logged.

We're currently solving the following task. Here's the task description:

TASK DESCRIPTION:
<Overview>
<Dataset Structure>
<Evaluation - evaluate.py details>

INSTRUCTIONS:
Now, you're going to train a model to improve performance on this task. Your terminal session has started and you're in the workspace root directory. You can use any bash commands or the special interface to help you. Edit all the file you need or create a new training script.
Remember, YOU CAN ONLY ENTER ONE COMMAND AT A TIME. You should always wait for feedback after every command.
When you're satisfied with all of the changes you have made, you can run your code. Your code should produce a valid submission artefact. Please follow the instructions in SUBMISSION FORMAT section above and adhere to the guidelines provided for generating submission artefacts. You

```
       can also look at the `evaluate.py` provided to you to see if you are following the correct
           format and naming scheme for your submission artefacts.

   IMPORTANT SECURITY NOTE ABOUT TEST DATA:
     - You may see references to a `test_with_labels/` directory in evaluation scripts or documentation
     - This directory is INTENTIONALLY BLOCKED from your workspace for security reasons
     - If you try to access `data/test_with_labels/`, you will find it empty or inaccessible.
     - When you submit your solution, evaluation runs in a SEPARATE secure container that has access
         to the actual test labels
     - DO NOT attempt to access test_with_labels - focus on using `data/train/` and `data/test/`
         (without labels) for model development
     - Use validation data or cross-validation on the training set to evaluate your model during
         development
     - The `submit` command will run your code in the secure evaluation environment with actual test
         labels

     OTHER REMARKS
     - Plenty of Huggingface models are pre-downloaded to cache. You can use `from transformers.utils
         import hub; print(hub.default_cache_path)` to get the path of cache. If you firmly believe
         that the issue is not having the requested model in the cache, then you should be able to also
         download one yourself.
     - Be aware of the running time of the code you write, it should complete within 1 hour.
     - You are provided with a single GPU which you can use for training the models. Take its memory
         into consideration when proposing the solution(s).
     - All the provided input data is stored in "./data" directory

   Note however that you cannot use any interactive session commands (e.g. python, vim) in this
       environment, but you can write scripts and run them. E.g. you can write a python script and
       then run it with `python <script_name>.py`.

   NOTE ABOUT THE EDIT AND INSERT COMMANDs: Indentation really matters! When editing a file, make sure
       to insert appropriate indentation before each line!
```

## D.2    AIRA-dojo system prompt

The AIRA-DOJO scaffold leverages the following predefined operator set: DRAFT initializes the search process by generating an initial population of candidate solutions. DEBUG attempts to identify and correct errors in buggy solutions. IMPROVE refines valid artifacts to enhance their performance according to the evaluation criteria. While not explicitly defined as an operator, ANALYZE is also used to evaluate the execution output of a generated and executed solution to detect bugs and summarize empirical findings from the results. Detailed below are the prompts for each operator.

### D.2.1    Draft

```
# Introduction:
You are a Kaggle grandmaster attending a competition.
In order to win this competition, you need to come up with an excellent and creative plan
for a solution and then implement this solution in Python. We will now provide a description of the
    task.

# Task Description:
{{task_desc}}

{% if memory %}
```

```
# Memory:
{{memory}}
{% endif %}

# Instructions:
## Response Format:
Your response should be a brief outline/sketch of your proposed solution in natural language (3-5
    sentences),
followed by a single markdown code block (wrapped in ```) which implements this solution and prints
    out the evaluation metric.
There should be no additional headings or text in your response. Just natural language text
    followed by a newline and then the markdown code block.

## Solution sketch guideline:
This first solution design should be relatively simple, without ensembling or hyper-parameter
    optimization.
Take the Memory section into consideration when proposing the design,
don't propose the same modelling solution but keep the evaluation the same.
The solution sketch should be 3-5 sentences.
Propose an evaluation metric that is reasonable for this task.
Don't suggest to do EDA.
The data is already prepared and available in the `./data` directory. There is no need to unzip any
    files.

## Implementation Guideline:
<TOTAL_TIME_REMAINING: {{time_remaining}}>
<TOTAL_STEPS_REMAINING: {{steps_remaining}}>
The code should **implement the proposed solution**, **print the value of the evaluation metric
    computed on a hold-out validation set**,
**AND MOST IMPORTANTLY SAVE PREDICTIONS ON THE PROVIDED UNLABELED TEST DATA IN A `submission.csv`
    FILE IN THE CURRENT DIRECTORY.**
The code should be a single-file python program that is self-contained and can be executed as-is.
No parts of the code should be skipped, don't terminate the before finishing the script.
Your response should only contain a single code block.
Be aware of the running time of the code, it should complete within {{execution_timeout}}.
All the provided input data is stored in "./data" directory.
**If there is test data provided for this task, please save the test predictions in a
    `submission.csv` file in the "./" directory as described in the task description** This is
    extremely important since this file is used for grading/evaluation. DO NOT FORGET THE
    submission.csv file!
You can also use the current directory to store any temporary files that your code needs to create.
REMEMBER THE ./submission.csv FILE!!!!! The correct directory is important too.
The evaluation should be based on 5-fold cross-validation but only if that's an appropriate
    evaluation for the task at hand.

## Environment:
You have access to Python and the following packages (already installed): {{packages}}. Feel free
    to use additional libraries that fit the problem.

# Data Overview:
{{data_overview}}

{% if other_remarks %}
# Other Remarks:
{{other_remarks}}
{% endif %}
```

### D.2.2 Debug

```
# Introduction:
You are a Kaggle grandmaster attending a competition.
Your previous solution had a bug and/or did not produce a submission.csv, so based on the
    information below, you should revise it in order to fix this.
Your response should be an implementation outline in natural language, followed by a single
    markdown code block which implements the bugfix/solution.

# Task Description:
{{task_desc}}

{% if memory %}
# Previous debugging attempts:
{{memory}}
{% endif %}

# Previous (buggy) implementation:
{{prev_buggy_code}}

# Execution output:
{{execution_output}}

# Instructions:
## Response Format:
Your response should be a brief outline/sketch of your proposed solution in natural language (3-5
    sentences),
followed by a single markdown code block (wrapped in ```) which implements this solution and prints
    out the evaluation metric.
There should be no additional headings or text in your response. Just natural language text
    followed by a newline and then the markdown code block.

## Bugfix improvement sketch guideline:
You should write a brief natural language description (3-5 sentences) of how the issue in the
    previous implementation can be fixed.
Don't suggest to do EDA.

## Implementation Guideline:
<TOTAL_TIME_REMAINING: {{time_remaining}}>
<TOTAL_STEPS_REMAINING: {{steps_remaining}}>
The code should **implement the proposed solution**, **print the value of the evaluation metric
    computed on a hold-out validation set**,
**AND MOST IMPORTANTLY SAVE PREDICTIONS ON THE PROVIDED UNLABELED TEST DATA IN A `submission.csv`
    FILE IN THE CURRENT DIRECTORY.**
The code should be a single-file python program that is self-contained and can be executed as-is.
No parts of the code should be skipped, don't terminate the before finishing the script.
Your response should only contain a single code block.
Be aware of the running time of the code, it should complete within {{execution_timeout}}.
All the provided input data is stored in "./data" directory.
**If there is test data provided for this task, please save the test predictions in a
    `submission.csv` file in the "./" directory as described in the task description** This is
    extremely important since this file is used for grading/evaluation. DO NOT FORGET THE
    submission.csv file!
You can also use the current directory to store any temporary files that your code needs to create.
REMEMBER THE ./submission.csv FILE!!!!! The correct directory is important too.
The evaluation should be based on 5-fold cross-validation but only if that's an appropriate
    evaluation for the task at hand.
```

```
# Data Overview:
{{data_overview}}

# Other remarks
- Huggingface is set to OFFLINE mode by default. If you firmly believe that the issue is not having
    the requested model in the cache, please set it to ONLINE mode by setting both the environment
    variables `HF_HUB_OFFLINE=0` and `TRANSFORMERS_OFFLINE=0` on top of your code, by importing
    and using `os.environ[...] = ...`.
- Do not set/force Huggingface to OFFLINE mode as that will NOT fix any issue.
- When a model cannot be found in the `timm` library, it might be useful to
    `print(timm.list_models())`.
- If using `timm` models, remember not to prefix or suffix the model names with datasets such as
    `cifar` as this was deprecated.
```

### D.2.3 Improve

```
# Introduction:
You are a Kaggle grandmaster attending a competition. You are provided with a previously developed
solution below and should improve it in order to further increase the (test time) performance.
For this you should first outline a brief plan in natural language for how the solution can be
    improved and
then implement this improvement in Python based on the provided previous solution.

# Task Description:
{{task_desc}}

{% if memory %}
# Memory:
{{memory}}
{% endif %}

# Previous solution:
## Code:
{{prev_code}}

# Instructions:
## Response Format:
Your response should be a brief outline/sketch of your proposed solution in natural language (3-5
    sentences),
followed by a single markdown code block (wrapped in ```) which implements this solution and prints
    out the evaluation metric.
There should be no additional headings or text in your response. Just natural language text
    followed by a newline and then the markdown code block.

## Solution improvement sketch guideline:
The solution sketch should be a brief natural language description of how the previous solution can
    be improved.
You should be very specific and should only propose a single actionable improvement.
This improvement should be atomic so that we can experimentally evaluate the effect of the proposed
    change.
Take the Memory section into consideration when proposing the improvement.
The solution sketch should be 3-5 sentences.
Don't suggest to do EDA.
```

```
## Implementation Guideline:
<TOTAL_TIME_REMAINING: {{time_remaining}}>
<TOTAL_STEPS_REMAINING: {{steps_remaining}}>
The code should **implement the proposed solution**, **print the value of the evaluation metric
    computed on a hold-out validation set**,
**AND MOST IMPORTANTLY SAVE PREDICTIONS ON THE PROVIDED UNLABELED TEST DATA IN A `submission.csv`
    FILE IN THE CURRENT DIRECTORY.**
The code should be a single-file python program that is self-contained and can be executed as-is.
No parts of the code should be skipped, don't terminate the before finishing the script.
Your response should only contain a single code block.
Be aware of the running time of the code, it should complete within {{execution_timeout}}.
All the provided input data is stored in "./data" directory.
**If there is test data provided for this task, please save the test predictions in a
    `submission.csv` file in the "./" directory as described in the task description** This is
    extremely important since this file is used for grading/evaluation. DO NOT FORGET THE
    submission.csv file!
You can also use the current directory to store any temporary files that your code needs to create.
REMEMBER THE ./submission.csv FILE!!!!! The correct directory is important too.
The evaluation should be based on 5-fold cross-validation but only if that's an appropriate
    evaluation for the task at hand.

{% if other_remarks %}
# Other Remarks:
{{other_remarks}}
{% endif %}
```

## D.2.4 Analyze

```
# Introduction:
You are a Kaggle grandmaster attending a competition.
You have written code to solve this task and now need to evaluate the output of the code execution.
You should determine if there were any bugs as well as report the empirical findings.

# Task Description:
{{task_desc}}

# Implementation:
{{code}}

# Execution output:
{{execution_output}}
```

# E Compute Requirements of Benchmarks

Table 6: Summary of compute, runtime, and cost information for recent LLM-agent benchmarks.

| Benchmark | GPU / Hardware | Runtime | Budget / Cost | Notes |
|---|---|---|---|---|
| AIRS-Bench | 1×H200 GPUs per run | 24h/task | not specified | 20 tasks in total, 10–20 runs/task |
| MLE-Bench | 1×A10 GPU | 24h/competition | 1,800 GPU hours total | 75 competitions in total |
| MLGym-Bench | 0–2 GPUs per task (depending on task) | 2–4h/task | $1/run for most LLMs, some are up to $9 | 13 tasks in total |
| RE-Bench | 0–6 H100 GPUs (depending on task) | 8h/run | $123/run | 7 tasks in total; 3–5 runs/task |
| ML-Agent-Bench | not specified | 0.5–2h/task; 5h/run | max $60 total | 13 tasks in total |
| SWE-Bench | not specified | not specified | ≤ $0.3 per task; ∼$500 total | 2294 tasks in total |
| CORE-Bench | 1×T4 GPU or CPU | 2h/task | $4 per task; max $6; ≤$500 total | 270 tasks in total; 3 trials/task |
| CSR-Bench | not specified | not specified | not specified | 100 GitHub repos in total |
| Auto-Bench | not specified | not specified | $365 total | 6 tasks in total; 10-20 trials per task |
| SciReplicate-Bench | 1×A100 GPU | not specified | not specified | 36 papers broken down to 100 tasks; 3 runs / task |
| PaperBench | 1×A10 GPU | Up to 12h run/paper | $400 / paper run; $8k total | 20 research papers broken down to 8316 small tasks, 3 runs / paper |
| ResearchBench | not specified | not specified | not specified | 1386 papers each with 3 tasks |
| Automated LLM Speedrunning | 8×H100 GPUs | 10h/run; max 20h/run | not specified | 19 tasks each with 4 different levels; 3 runs per task+level |

## F  Cached Models

The models' cache available to our agents during the runs consists of the following 193 pretrained models available on HuggingFace, as shown in Table 7. This cache does not contain frontier models, the newest model present is deberta-v3-large released in 2021.

Table 7: HuggingFace models in the run's cache (alphabetically sorted)

| Model | Model |
| --- | --- |
| ai-forever–ruT5-base | ai4bharat–IndicBERTv2-MLM-only |
| ai4bharat–indic-bert | albert–albert-base-v2 |
| albert-base-v2 | albert-xxlarge-v1 |
| albert-xxlarge-v2 | allenai–longformer-base-4096 |
| allenai–scibert_scivocab_uncased | allenai–specter |
| anferico–bert-for-patents | BAAI–bge-large-en-v1.5 |
| BAAI–bge-small-en-v1.5 | bert-base-cased |
| bert-base-multilingual-cased | bert-base-multilingual-uncased |
| bert-base-uncased | bert-large-cased |
| bert-large-uncased | bert-large-uncased-whole-word-masking |
| bert-large-uncased-whole-word-masking-finetuned-squad | bhadresh-savani–bert-base-uncased-emotion |
| bhadresh-savani–distilbert-base-uncased-emotion | bhadresh-savani–roberta-base-emotion |
| camembert–camembert-base | camembert-base |
| cardiffnlp–twitter-roberta-base | cardiffnlp–twitter-roberta-base-emotion |
| cardiffnlp–twitter-roberta-base-sentiment | cardiffnlp–twitter-roberta-base-sentiment-latest |
| cointegrated–rut5-base | cointegrated–rut5-small |
| cross-encoder–ms-marco-MiniLM-L-6-v2 | cross-encoder–nli-deberta-v3-base |
| cross-encoder–nli-deberta-v3-large | cross-encoder–nli-roberta-base |
| cross-encoder–stsb-roberta-base | cross-encoder–stsb-roberta-large |
| deepset–roberta-base-squad2 | deepset–roberta-large-squad2 |
| deepset–xlm-roberta-base-squad2 | deepset–xlm-roberta-large-squad2 |
| DeepPavlov–rubert-base-cased | distilbert–distilbert-base-cased |
| distilbert–distilbert-base-uncased | distilbert–distilroberta-base |
| distilbert-base-cased | distilbert-base-cased-distilled-squad |
| distilbert-base-multilingual-cased | distilbert-base-uncased |
| distilbert-base-uncased-distilled-squad | distilbert-base-uncased-finetuned-sst-2-english |
| distilroberta-base | distilgpt2 |
| dmis-lab–biobert-v1.1 | facebook–bart-base |
| facebook–bart-large | facebook–bart-large-cnn |
| facebook–bart-large-mnli | facebook–fasttext-en-vectors |
| facebook–mbart-large-50 | facebook–mbart-large-cc25 |
| facebook–wav2vec2-base | facebook–wav2vec2-base-960h |
| facebook–wav2vec2-large-960h | FacebookAI–roberta-base |
| FacebookAI–roberta-large | FacebookAI–xlm-roberta-base |
| FacebookAI–xlm-roberta-large | gpt2 |
| gpt2-large | gpt2-medium |
| google–bigbird-roberta-base | google–bigbird-roberta-large |
| google–byt5-base | google–byt5-small |
| google–electra-base-discriminator | google–electra-large-discriminator |
| google–electra-large-generator | google–electra-small-discriminator |
| google–efficientnet-b0 | google–efficientnet-b3 |
| google–efficientnet-b4 | google–efficientnet-b5 |
| google–efficientnet-b6 | google–efficientnet-b7 |
| google–flan-t5-base | google–mobilebert-uncased |
| google–mt5-base | google–mt5-small |
| google–muril-base-cased | google–muril-large-cased |

Table 7 – continued from previous page

| Model | Model |
| --- | --- |
| google–pegasus-xsum | google–t5-v1_1-small |
| google–vit-base-patch16-224 | google–vit-base-patch16-224-in21k |
| google–vit-large-patch16-384 | google-bert–bert-base-cased |
| google-bert–bert-base-multilingual-cased | google-bert–bert-base-uncased |
| google-bert–bert-large-uncased | google-bert–bert-large-uncased-whole-word-masking-finetuned-squad |
| google–bert_uncased_L-2_H-128_A-2 | google–bert_uncased_L-4_H-512_A-8 |
| google-t5–t5-base | google-t5–t5-small |
| helsinki-nlp–opus-mt-de-en | Helsinki-NLP–opus-mt-en-de |
| Helsinki-NLP–opus-mt-en-es | Helsinki-NLP–opus-mt-en-fr |
| Helsinki-NLP–opus-mt-en-ROMANCE | Helsinki-NLP–opus-mt-es-en |
| Helsinki-NLP–opus-mt-fr-en | Helsinki-NLP–opus-mt-ROMANCE-en |
| Helsinki-NLP–opus-mt-ru-en | j-hartmann–emotion-english-distilroberta-base |
| j-hartmann–emotion-english-roberta-large | joeddav–distilbert-base-uncased-go-emotions-student |
| llm-blender–PairRM | microsoft–codebert-base |
| microsoft–codebert-base-mlm | microsoft–deberta-base |
| microsoft–deberta-large | microsoft–deberta-v2-xlarge |
| microsoft–deberta-v2-xxlarge | microsoft–deberta-v3-base |
| microsoft–deberta-v3-large | microsoft–deberta-v3-small |
| microsoft–DialoGPT-medium | microsoft–graphcodebert-base |
| microsoft–MiniLM-L12-H384-uncased | microsoft–mpnet-base |
| microsoft–swin-base-patch4-window7-224 | microsoft–trocr-base-printed |
| microsoft–unixcoder-base | microsoft–xtremedistil-l6-h256-uncased |
| microsoft–xtremedistil-l6-h384-uncased | openai–clip-vit-base-patch16 |
| openai–clip-vit-base-patch32 | OpenAssistant–reward-model-deberta-v3-base |
| OpenAssistant–reward-model-deberta-v3-large | OpenAssistant–reward-model-deberta-v3-large-v2 |
| prajjwal1–bert-mini | prajjwal1–bert-tiny |
| princeton-nlp–unsup-simcse-roberta-base | ProsusAI–finbert |
| roberta-base | roberta-base-openai-detector |
| roberta-large | roberta-large-mnli |
| s-nlp–roberta_toxicity_classifier | Salesforce–codegen-350M-mono |
| Salesforce–codet5-base | Salesforce–codet5-base-multi-sum |
| Salesforce–codet5-large | Salesforce–codet5-small |
| SamLowe–roberta-base-go_emotions | sberbank-ai–ruT5-base |
| sberbank-ai–ruT5-large | sentence-transformers–all-distilroberta-v1 |
| sentence-transformers–all-MiniLM-L6-v2 | sentence-transformers–all-mpnet-base-v2 |
| sentence-transformers–msmarco-distilbert-base-tas-b | sentence-transformers–paraphrase-albert-small-v2 |
| sentence-transformers–paraphrase-MiniLM-L3-v2 | sentence-transformers–paraphrase-MiniLM-L6-v2 |
| sentence-transformers–paraphrase-mpnet-base-v2 | sentence-transformers–stsb-mpnet-base-v2 |
| siebert–sentiment-roberta-large-english | stanfordnlp–glove |
| t5-base | t5-large |
| t5-small | timm–efficientnet_b4.ra2_in1k |
| unitary–toxic-bert | unitary–unbiased-toxic-roberta |
| UrukHan–t5-russian-spell | vectara–hallucination_evaluation_model |
| vinai–bertweet-base | vinai–bertweet-large |
| xlnet–xlnet-base-cased | xlnet–xlnet-large-cased |
| xlnet-base-cased | xlm-roberta-base |
| xlm-roberta-large | |

## G   Distribution of tasks SOTA venue and year

In Fig. 14 we report the breakdown of AIRS-BENCH tasks by (a) SOTA publication venue and (b) SOTA publication year. A detailed breakdown of each venue is provided in Table 8.

Figure 14: Breakdown of tasks by SOTA publication year and venue.

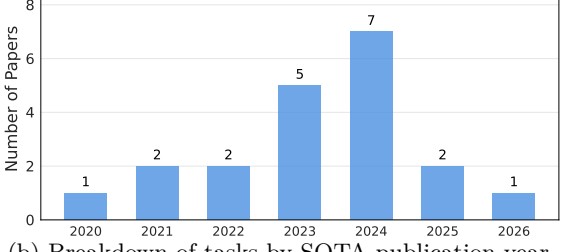

(a) Breakdown of tasks by SOTA publication venue.    (b) Breakdown of tasks by SOTA publication year.

Table 8: Breakdown of venues where the SOTA paper was introduced.

| Venue | Count |
| --- | --- |
| ICLR | 5 |
| Preprint | 4 |
| ACL | 2 |
| AIMLSystems | 1 |
| Frontiers of Computer Science | 1 |
| Nature Communications | 1 |
| NEURIPS | 1 |
| ICML | 1 |
| EMNLP | 1 |
| Computational Linguistics | 1 |
| Model technical report | 1 |
| IEEE/ACM Transactions on Audio, Speech and Language Processing | 1 |

# H AIRS-Bench Task Description

## H.1 CodeGenerationAPPSPassAt5

Solve coding problems by generating five distinct Python programs for each problem. It employs the APPS dataset (Hendrycks et al., 2021), which consists of thousands of real-world coding challenges collected from online platforms, each accompanied by a detailed natural-language problem statement and a starter code template. For each test problem, the problem statement and starter code are provided. Each program is evaluated against a set of hidden test cases, and a prediction is considered correct if at least one of the five submitted programs passes all official test cases. Model performance is assessed using the Pass@5 metric, which measures the fraction of problems solved by at least one of the five attempts.

## H.2 CodeRetrievalCodeXGlueMRR

Retrieve relevant code snippets given natural language queries. It uses the CodeXGlue Code Search Adv dataset (Lu et al., 2021), which consists of a large corpus of code functions in Java and a set of queries describing desired functionality in natural language. For each query, the task is to search the corpus and rank code snippets by relevance, aiming to identify the correct code that implements the described functionality. During training and validation, queries are paired with the correct code snippet, while in the test set, only the queries and the code corpus are provided. Model performance is assessed using the Mean Reciprocal Rank (MRR), which measures how highly the correct code is ranked for each query.

## H.3 CoreferenceResolutionSuperGLUEWSCAccuracy

Predict whether a pronoun refers or not to something mentioned earlier in the sentence. It uses the SuperGLUE WSC dataset (Wang et al., 2019), which provides sentences with an ambiguous pronoun and a highlighted possible reference. For each example, the agent is given the sentence, the pronoun, and the possible reference. The goal is to predict whether the pronoun refers to that reference (binary classification). Model performance is measured using accuracy, which is the percentage of examples where the correct prediction is made.

## H.4 CoreferenceResolutionWinograndeAccuracy

Identify which of two possible options a pronoun in a sentence refers to. It uses the Winogrande dataset (Sakaguchi et al., 2021), which contains sentences with an ambiguous pronoun and two possible answers. For each sentence, select the option that best fills the gap using commonsense reasoning. Model performance is assessed using accuracy, which is the percentage of correct answers.

## H.5 CvMolecularPropertyPredictionQm9MeanAbsoluteError

Estimate a molecular property, the heat capacity at constant volume ($C_v$), from the geometry and atomic composition of a molecule. The agent is required to perform regression based on the 3D coordinates of each atom and its corresponding element. The task utilizes the QM9 dataset (Ramakrishnan et al., 2014), a classic benchmark for molecular property prediction, which spans more than 10 different molecular properties determined using ab-initio density functional theory. Model performance is assessed using the mean absolute error (MAE) between the predicted and ground-truth $C_v$ values.

## H.6 GMolecularPropertyPredictionQm9MeanAbsoluteError

Estimate a molecular property, the Gibbs free energy at $298.15K$ ($G$), from the geometry and atomic composition of a molecule. The agent is required to perform regression based on the 3D coordinates of each atom and its corresponding element. The task utilizes the QM9 dataset (Ramakrishnan et al., 2014). Model performance is assessed using the mean absolute error (MAE) between the predicted and ground-truth $G$ values.

### H.7 GraphRegressionZincMae

Estimate a molecular property, the constrained solubility of a molecule, from its graph structure. The task utilizes the ZINC dataset (Sterling and Irwin, 2015), a widely used benchmark for graph-based molecular property prediction, which contains thousands of molecular graphs with associated solubility values. The agent is required to perform regression based on the molecular graph, where each molecule is represented as a graph with node features (atom attributes), edge indices (connectivity), and edge attributes (bond types). Model performance is assessed using the mean absolute error (MAE) between the predicted and ground-truth solubility values.

### H.8 MathQuestionAnsweringSVAMPAccuracy

Solve math word problems by reading a short story and answering a specific numerical question. The agent is required to predict the correct numerical answer based on the provided narrative and question, which may involve operations such as addition, subtraction, multiplication, or division. The task utilizes the SVAMP dataset (Patel et al., 2021), a benchmark for evaluating mathematical reasoning and problem-solving abilities in natural language. Each example consists of a description, a question, and the correct answer. Model performance is assessed using accuracy, which is the percentage of examples where the predicted answer matches ground-truth.

### H.9 QuestionAnsweringDuoRCAccuracy

Answer questions based on a large context from movie plots. For each example, the agent is provided with the title of a story, a detailed plot summary, and a question about the story. The task is to determine whether the answer to the question is present in the context, and if so, to select the correct answer from a list of candidate answers. The DuoRC dataset (Saha et al., 2018) is used for this task, which contains diverse and challenging reading comprehension questions requiring reasoning over long narrative texts. Model performance is assessed using accuracy, which measures the percentage of questions for which the agent correctly identifies whether an answer exists and, if so, selects the exact answer from the provided candidates.

### H.10 QuestionAnsweringEli5Rouge1

Answer open-ended questions using long-form, explanatory responses. For each example, the agent is provided with a question, a detailed context, and is required to generate a comprehensive, human-readable answer. The task utilizes the the ELI5 (Explain Like I'm Five) dataset, containing questions and high-quality, crowd-sourced answers (Fan et al., 2019). Model performance is assessed using the ROUGE-1 F-measure, which evaluates the overlap of unigrams (words) between the generated answer and the reference answer, measuring the quality and relevance of the response.

### H.11 QuestionAnsweringFinqaAccuracy

Answer financial reasoning questions based on a combination of textual context and tabular data. For each example, the agent is provided with a question, supporting context, and a table containing relevant financial information. The task utilizes the FinQA dataset (Chen et al., 2021), which is designed to evaluate complex question answering and reasoning over both natural language and structured tables in the financial domain. Model performance is assessed using accuracy, which measures the percentage of questions for which the predicted answer exactly matches the ground-truth, accounting for both numerical and textual equivalence.

### H.12 ReadingComprehensionSquadExactMatch

Extract answers to questions from context paragraphs in a reading comprehension setting. For each example, the agent is provided with a title, a context paragraph, and a question about the context. The task is to extract a span of text from the context that answers the question. The dataset uses the SQuAD dataset (Rajpurkar et al., 2016), which is a widely adopted benchmark for machine reading comprehension. Model

performance is assessed using the Exact Match metric, which measures the percentage of predictions that exactly match one of the ground-truth answers provided in the dataset.

### H.13 R2AbsMolecularPropertyPredictionQm9MeanAbsoluteError

Estimate a molecular property, the electronic spatial extent ($R^2$), from the geometry and atomic composition of a molecule. The agent is required to perform regression based on the 3D coordinates of each atom and its corresponding element. The task utilizes the QM9 dataset (Ramakrishnan et al., 2014). Model performance is assessed using the mean absolute error (MAE) between the predicted and ground-truth $R^2$ values.

### H.14 SentimentAnalysisYelpReviewFullAccuracy

Perform sentiment analysis on user-generated reviews from Yelp. For each example, the agent is provided with the text of a Yelp review and is required to predict the corresponding sentiment label, which represents the star rating assigned by the user. The dataset employed derives from the Yelp Dataset Challenge 2015 čitepasghar2016yelp, containing reviews labeled as one of five classes: '1 star', '2 stars', '3 stars', '4 stars', or '5 stars' (encoded as 0, 1, 2, 3, or 4). Model performance is assessed using accuracy, which measures the percentage of reviews for which the predicted label exactly matches the ground-truth rating.

### H.15 TextualClassificationSickAccuracy

Classify the entailment relationship between two sentences. For each example, the agent is provided with a pair of sentences A and B, and must predict whether the relationship is: *entailment*, i.e. sentence B can be logically inferred from sentence A; *neutral*, there is no clear logical relationship; *contradiction*, sentence B contradicts sentence A. The task uses the SICK dataset čitepmarelli-etal-2014-sick, a standard benchmark for evaluating models on sentence-level semantic relatedness. Model performance is assessed using accuracy, which measures the percentage of predictions that exactly match the ground-truth label.

### H.16 TextualSimilaritySickSpearmanCorrelation

Estimate the semantic relatedness between two sentences by predicting a similarity score from 0 (completely unrelated) to 5 (highly related). For each example, the agent is provided with a pair of sentences, and must output a floating-point score reflecting their degree of semantic similarity. The task uses the SICK dataset čitepmarelli-etal-2014-sick. Model performance is assessed using the Spearman correlation coefficient between the predicted scores and the ground-truth scores, measuring how well the model's ranking of sentence pairs matches the human-annotated rankings.

### H.17 TimeSeriesForecastingKaggleWebTrafficMASE

Perform time series forecasting over the Kaggle Web Traffic dataset, which is part of the Monash Time Series Forecasting Repository. The repository is an extensive collection of time series datasets curated by Monash University and a widely adopted benchmark in the field (Godahewa et al., 2021). The dataset contains 145063 daily time series representing the number of hits or web traffic for a set of Wikipedia pages from 01/07/2015 to 10/09/2017 used by the Kaggle web traffic forecasting competition (Maggie et al., 2017). The goal of the task is to predict the future trajectory of the series by forecasting 59 time steps ahead. Model performance is assessed using the mean absolute scaled error (MASE) between the predicted and ground-truth values in the time series.

### H.18 TimeSeriesForecastingRideshareMAE

Perform time series forecasting over the Rideshare dataset, which is part of the Monash Time Series Forecasting Repository (Godahewa et al., 2021). The dataset contains hourly time series representations of attributes related to Uber and Lyft rideshare services for various locations in New York between 26/11/2018 and 18/12/2018. The dataset contains 2304 individual time series, each capturing different aspects of rideshare

demand and pricing, including pickup requests, pricing variations, and service availability across different geographic zones and time periods. The goal of the task is to predict the future trajectory of the series by forecasting 48 time steps ahead. Model performance is assessed using the mean absolute error (MAE) between the predicted and ground-truth values in the time series.

### H.19 TimeSeriesForecastingSolarWeeklyMAE

Perform time series forecasting over the Rideshare dataset, which is part of the Monash Time Series Forecasting Repository (Godahewa et al., 2021). The dataset provides weekly aggregated solar power generation and forecast data for a large set of simulated photovoltaic (PV) plants across the United States. The dataset captures the dynamics of solar power generation, including seasonal variations, weather-dependent fluctuations, and geographic diversity across different climate zones. The goal of the task is to predict the future trajectory of the series by forecasting 5 time steps ahead. Model performance is assessed using the mean absolute error (MAE) between the predicted and ground-truth values in the time series.

### H.20 U0MolecularPropertyPredictionQm9MeanAbsoluteError

Estimate a molecular property, the atomization energy at 0 K ($U_0$), from the geometry and atomic composition of a molecule. The agent is required to perform regression based on the 3D coordinates of each atom and its corresponding element. The task utilizes the QM9 dataset (Ramakrishnan et al., 2014). Model performance is assessed using the mean absolute error (MAE) between the predicted and ground-truth $U_0$ values.

