# OpenReview forum: "AIRS-Bench: A Suite of Tasks for Frontier AI Research Science Agents"
_TMLR — Under review for TMLR_

### Review · Reviewer_T3wr · 2026-06-24

**Summary Of Contributions:**

This paper introduces AIRS-Bench, a benchmark for evaluating LLM-based agents on machine-learning research tasks. The benchmark contains 20 tasks derived from recent ML papers and covers NLP, code, math, molecular property prediction, and time-series forecasting. Each task provides a problem description, data preparation code, an evaluation script, and metadata. The agent is expected to write code, train or adapt a model, and submit predictions for hidden-label evaluation.

The paper also defines an evaluation protocol with valid submission rate, normalized score, and Elo-style pairwise ratings. The experiments cover 27 LLM–scaffold configurations, including one-shot, greedy/tree-search, and ReAct-style agents. The main empirical results are that greedy/tree-search agents usually perform better than one-shot agents, that valid submission is still a failure mode for many configurations, and that some agents exceed the reported literature SOTA on 7 of the 20 tasks.

The main strengths are the standardized task packaging, the use of hidden test labels, the multi-seed evaluation, and the attempt to compare several agent/scaffold configurations under a shared benchmark format. The main weaknesses are that several central claims are stronger than the current evidence supports. In particular, the paper does not yet justify the term “uncontaminated,” the comparison to “human SOTA” is not always controlled for resources and model access, the normalized score depends on the set of evaluated agents, and the scaffold comparison is partly confounded with harness differences.

**Additional Comments:**

The paper would benefit from more conservative wording throughout. I would avoid “uncontaminated,” “human SOTA,” and “full research lifecycle” unless each term is defined and supported by the protocol. Phrases such as “reported reference score,” “hidden-label no-starter-code setting,” and “ML engineering workflow” would be more precise in several places.

There are also some smaller issues to fix before publication. The paper should report exact closed-model versions and access dates. The comparison table with related benchmarks is useful, but some entries, such as scientific-method stages and baseline access, are partly subjective and should be briefly justified. The appendix is useful, but several task descriptions and citation artifacts need proofreading.

**Audience:**

Yes

**Audience Explanation:**

This paper introduces AIRS-Bench, a benchmark for evaluating LLM-based agents on machine-learning research tasks.

**Broader Impact Concerns:**

A Broader Impact Statement should be added. The paper evaluates agents that can automate parts of the ML research workflow, including code generation, model training, debugging, and submission to hidden-label evaluations. This has potential benefits for research productivity, but it also raises concerns.

First, such agents could accelerate poorly validated model development if benchmark progress is treated as a substitute for scientific judgment. Second, several tasks involve molecular or biochemical modeling. The current tasks appear to use public ML datasets rather than wet-lab protocols, but the paper should still state how dual-use scientific tasks were considered and whether any tasks were excluded. Third, the benchmark uses long GPU runs and many seeds, so cost and access may limit who can reproduce or extend the results.

The broader-impact discussion should cover sandboxing, hidden labels, restrictions on task selection, logging of external resources, compute cost, and the intended use of AIRS-Bench. I do not see a broader-impact concern that blocks publication by itself, but the current manuscript should address these points directly.

**Claims And Evidence:**

No

**Claims Explanation:**

The paper addresses a relevant evaluation problem, but the current version needs changes before I would recommend acceptance. The benchmark format is useful: the tasks are closer to applied ML workflows than short-form coding or QA benchmarks, and the paper reports more than a single leaderboard score. The empirical section also gives a reasonable first view of how current agents fail, especially through invalid submissions and weak performance on harder tasks.

The main issue is that the paper sometimes treats design choices as stronger evidence than they provide. For example, the benchmark hides test labels and does not give starter code, but that does not establish that the benchmark is uncontaminated. The source papers, datasets, public leaderboards, and possibly related repositories may still be present in pretraining data or reachable through web access. The paper should separate “no baseline code is provided” from broader contamination claims.

The SOTA comparison also needs more care. Some reported SOTA-beating cases may be valid and interesting, but the current wording can suggest that agents outperform human research methods under comparable conditions. In several cases, the agent may use newer pretrained models, larger backbones, ensembling, task-specific fine-tuning, or different compute than the referenced SOTA method. The paper should make those differences explicit and avoid using “human SOTA” as a single category unless the comparison protocol is matched.

The aggregate scoring protocol is another concern. The normalized score uses the worst observed score among the evaluated agents and seeds as a task-level anchor. This means the score is not fixed by the task alone; it depends on which agents were included in the evaluation. That makes future comparisons harder, because adding a weaker or different agent could change the normalization for existing results.

I am also concerned about the task selection procedure. The final 20 tasks are selected from a larger pool partly to preserve agent scores and rankings. That is a reasonable way to reduce compute, but it also means the benchmark may be fitted to the current agent set. The paper should either evaluate on a held-out task subset or present this as a limitation.

Finally, the scaffold comparison is not fully isolated. AIRA-dojo Greedy and MLGym ReAct differ in validation setup, scoring of intermediate solutions, time per solution, prompt/interface details, and harness behavior. The paper can still report these results, but claims about scaffold quality should be phrased as comparisons between evaluated harness–scaffold configurations unless a more controlled experiment is added.

**Requested Changes:**

- The paper should define what kind of contamination it rules out. At present, the evidence seems to support a narrower claim: agents do not receive baseline code, test labels, or the SOTA score in the task prompt. This does not rule out exposure through pretraining, public datasets, PapersWithCode, GitHub repositories, arXiv papers, or web access during the run. The paper should either provide a stronger contamination audit or replace “uncontaminated” with a narrower term such as “hidden-label” or “no-starter-code.”
- For each task, please report whether the dataset, task name, source paper, leaderboard entry, public code, and test split are publicly available. The paper should also state whether agents could access arXiv, PapersWithCode, GitHub, HuggingFace model pages, and task-specific repositories during execution. If web access is allowed, the authors should log or summarize external resources accessed by the agents. This is necessary to interpret whether agents are developing solutions or recovering known ones.
- The paper should provide a task-level table with the exact SOTA source, year, split, metric implementation, score, and whether the score was reproduced or only taken from the paper. For the seven tasks where an agent exceeds the reported SOTA, the authors should state whether the agent used newer pretrained models, larger backbones, additional data, ensembling, test-time augmentation, or more compute than the SOTA method. If the protocols differ, the result should be described as exceeding the reported reference score rather than beating human SOTA.
- The current score uses the worst observed score across agents and seeds as one anchor. This makes the normalized score depend on the agents included in the paper. The authors should use fixed task-level anchors, report raw scores more prominently, or show that rankings are stable under several normalization choices. The current normalization can remain as an analysis tool, but it should not be the only aggregate score for long-term benchmark comparison.
- The paper should give worked examples for both higher-is-better metrics and lower-is-better metrics. This is especially important for regression tasks such as MAE and MASE, where the optimum is 0 and the behavior of the log-distance transform is less intuitive than for accuracy.
- The final benchmark is selected from a larger task pool partly to preserve agent rankings and average scores. The paper should state whether the same agents used in the final evaluation were also used during task selection. If so, this should be treated as a limitation. A stronger version would reserve some tasks for final evaluation only, or at least show that the main conclusions hold under alternative 20-task subsets.
- The comparison between AIRA-dojo Greedy and MLGym ReAct changes more than the scaffold. The validation split, solution scoring policy, time per solution, interface, and harness behavior also differ. The authors should either run a controlled comparison in one harness or describe the result as a comparison between harness–scaffold configurations rather than scaffold designs alone.
- Since this is a benchmark paper, the release should include task definitions, evaluation scripts, prompts, container specifications, package versions, exact model names, API/model access dates, cached model revisions, raw per-seed results, invalid-submission logs, and scripts for normalized score and Elo computation. The paper should also report approximate GPU hours and monetary cost for the main experiments.
- The manuscript includes a GitHub URL that points to a non-anonymous repository. During double-blind review, this should be replaced by an anonymized repository or removed from the submitted version. The public repository can be restored after review.
- The paper evaluates agents that automate parts of ML research. That raises issues around compute use, uneven access to the benchmark, automated model development, and possible dual-use concerns for scientific modeling tasks. These issues should be discussed explicitly.
- The tasks require agents to write code, train or adapt models, debug, evaluate, and submit predictions. The evidence for literature review, problem formulation, and scientific hypothesis generation is weaker. The paper should describe the evaluated workflow more precisely instead of using broad “full lifecycle” wording.
- Treating invalid submissions as zero is reasonable for one aggregate score, because invalid outputs are a real failure mode. However, it combines two effects: the ability to submit a valid artifact and the quality of the artifact once valid. Please report normalized score both with invalid runs counted as zero and conditional on valid submissions.
- Figures 9 and 10 contain useful information, but they are difficult to read. A compact table with raw task-level scores for the main agents, the reported SOTA, and simple baselines would make the results easier to interpret.
- If the benchmark will be extended over time, the paper should explain how new tasks will be added, how old scores will remain comparable, and how normalization will be handled. If the benchmark is static, the paper should discuss leakage and saturation once the task definitions and evaluation scripts are public.
- A few standard baselines would help calibrate task difficulty. For example, linear models, small pretrained-model fine-tuning, or task-specific standard models would show how much improvement comes from agentic search rather than from using strong pretrained components.
- Some appendix task descriptions and citations need editing. For example, the SolarWeekly task appears to be described as using the Rideshare dataset, and some citation commands appear as unresolved LaTeX artifacts. These issues do not change the main results, but they matter for a benchmark paper because users will rely on the task descriptions.

---

### Review · Reviewer_4zkn · 2026-06-25

**Summary Of Contributions:**

The paper introduces a novel benchmark designed to rigorously evaluate the autonomous scientific research capabilities of large language model agents in machine learning. The benchmark comprises 20 uncontaminated, end-to-end tasks sourced from state-of-the-art literature across seven diverse domains. The experimental results reveal that while top-performing agents manage to surpass the human state-of-the-art (SOTA) in 7 tasks.

Stengths:

1. By withholding baseline starter code, the benchmark forces agents to explore and generate code entirely from scratch. This effectively mitigates the issue of data contamination (where LLMs memorize benchmark solutions from pre-training data) and tests true reasoning capabilities.

2. The implementation of Valid Submission Rates, non-linear normalized scores (using the "March of 9s" transform), and Elo ratings enables fair, aggregated, and statistically robust comparisons across heterogeneous tasks with vastly different metrics and scales.

**Audience:**

Yes

**Audience Explanation:**

Yes, the TMLR audience would be highly interested in these findings. The paper introduces a much-needed, standardized benchmark to rigorously evaluate the end-to-end capabilities of autonomous AI research agents.

**Broader Impact Concerns:**

None.

**Claims And Evidence:**

Yes

**Claims Explanation:**

Yes, the claims are well-supported by convincing evidence. The authors conducted extensive empirical evaluations, and they employed robust, clear metrics such as Elo ratings and non-linear normalized scores to validate their findings.

**Requested Changes:**

1. The AIRA-DOJO framework includes distinct operators like Draft, Debug, Improve, and Analyze. If a model performs exceptionally well under Greedy search, which specific step played the decisive role? If the Improve operator is removed, how much does the performance drop? Without such ablation studies, it is impossible to know exactly which parts of the scaffold design are genuinely effective.

2. The paper enforces a strict 24-hour limit per run. But is an agent's performance linearly correlated with its runtime? Or does it converge by the 4th hour? They should have included an experiment showing the performance curve over time (e.g., at 1h, 6h, 12h, and 24h marks). This would provide highly instructive guidance for researchers operating with limited compute resources.

---

### Review · Reviewer_hBvB · 2026-07-10

**Summary Of Contributions:**

The paper proposes AIRS-Bench, a benchmark suite for evaluating LLM-based agents on AI research tasks. The benchmark contains 20 tasks, spanning 7 distinct task categories. Comparing to existing popular agentic AI research benchmarks, AIRS-Bench (1) tackles long-horizon research challenges; (2) involves the complete research pipeline; (3) does not provide any baseline code; and (4) leverages enough computational resources. The paper also proposes a standardized task format, which makes the benchmark extensible and compatible with multiple agent harnesses. A large set of LLM-scaffold combinations are evaluated using valid submission rate, normalized score, and Elo rating. The empirical results suggest that current frontier agents can exceed the reported human SOTA on some tasks, but remain below SOTA on most tasks, indicating that the benchmark is not saturated.

Strengths:
1. The paper aims to tackle an important problem: how to evaluate agents that aim to automate parts of ML research.
2. Comparing to existing benchmarks, the proposed benchmark extends the evaluation across several dimensions: task composition and origin, reasoning horizon, the stages of the scientific method covered, access to baseline solutions, and compute requirements.
3. The paper is relatively comprehensive in its empirical evaluation, covering multiple frontier models, scaffolds, repeated seeds, and aggregate metrics.

Weaknesses:
1. Whether the benchmark is uncontaminated is questionable. The authors claim that the proposed benchmark is uncontaminated. However, the tasks are sourced from public papers and datasets. Agents may access pretrained models and the internet. The paper needs a much more careful analysis and clarification on what is the scope of uncontaminated, and why it is uncontaminated.
2. The comparison between scaffolds is difficult to interpret because scaffold choice is partially entangled with harness choice. For example, Greedy is evaluated through AIRA-dojo while ReAct is evaluated through MLGym.

**Audience:**

Yes

**Audience Explanation:**

The paper would potentially benefit the researchers in LLM agents and auto-research agents.

**Broader Impact Concerns:**

Given that the work explicitly aims to benchmark and accelerate AI research agents, the authors may discuss possible societal and community concerns, including benchmark gaming and risks of automated low-quality research production.

**Claims And Evidence:**

No

**Claims Explanation:**

Mostly yes, but the major concern is in Weakness 1. The tasks are derived from public ML papers and datasets, all of which may be present in model pretraining corpora. Also, the agents appear to have access to pretrained checkpoints and internet access. The authors need better explanation and analysis on the term “uncontaminated”.

**Requested Changes:**

1. See weakness 1. Clarify and substantiate the “uncontaminated” claim.
2. See weakness 2. Please consider disentangling the scaffold effects from harness/environment effects. If disentangling is not practical, please consider discussing the differences, such as validation split strategy, library versions, evaluation strategies etc.
3. Please provide a table for all 20 tasks specifying the exact split, metric, SOTA source, SOTA year, etc. This will be helpful for more clear comparison.
4. Clarify the definition of "human SOTA". In many benchmark contexts, “human SOTA” may refer to a reported result from a particular paper under specific evaluation protocol, not the current best human-designed method under an identical evaluation protocol.
5. Please include a section discussing broader impact concerns. See the section below.

---

### Review · Reviewer_eV7F · 2026-07-17

**Summary Of Contributions:**

The paper proposes a dataset that aims to provide a standardized environment to evaluate the research capability of LLM and scaffold based agent. The dataset contains a combination of 20 tasks spanning 7 categories, taken from SOTA papers published on conferences or ArXiV. The benchmark provides no baseline methods to the agents, and uses three metrics (validation submission rate, normalized score, and Elo rating system) to evaluate the performance of the agents. Based on the benchmark, the paper evaluated 27 agent configurations that involves SOTA models, and identified that in 7 out of the 20 tasks some agents exceeded the best human solutions.

**Strengths**
1. The tasks involved in the dataset comes from convincing source. The environment provided to the agent seems standardized.
2. The evaluation design involves three aspects, which is quite comprehensive.
3. The paper evaluated 27 agent configuration with SOTA performance on the benchmark.

**Weaknesses**
1. It is not clear whether the current design is really uncontaminated. Although the baseline method is not provided to the agent, it cannot be guaranteed whether the agent has seen the baseline method before or have access to it. Also, the evaluate.py script and utils.py, when provided to the agent, may leak task-related info to the agents.
2. The normalization in the normalized score is not so convincing, as it depends on the worst performing solution. In particular, if there is an extremely bad submission, then all the agent's normalized score will be pulled to very close to 1, influencing the significance of the advantange of some agents.
3. It is not clear to me why the propose benchmark provides a full research lifecycle, and why the other benchmarks do not. Maybe a clearer explanation is needed.
4. In terms of task coverage, it seems that the image-based tasks are explicitly excluded. Not sure whether this is the case.
5. The paper argues in the conclusion that "For most tasks, even the best performing agent is still significantly behind the human SOTA, showing that the benchmark is far from saturated." It might be unfair to compare the agents with a limited running time and the human SOTA, which might be developed over months to years of research.

**Audience:**

Yes

**Audience Explanation:**

Agents are getting more and more capable for conducting scientific research. It is important to properly evaluate their capabilities in this perspective.

**Claims And Evidence:**

No

**Claims Explanation:**

See Weaknesses #1 and #3

**Requested Changes:**

1. Provide a more thorough explanation of why the tasks are uncontaminated.
2. Provide an explanation of why the propose benchmark provides a full research lifecycle, and why the other benchmarks do not.
3. Discuss more thoroughly about the comparison between the agent's solution and human SOTA solutions.

---

### Review · Reviewer_R38c · 2026-07-20

**Summary Of Contributions:**

The work introduces AIRS-Bench, a benchmark of 20 long-horizon, end-to-end AI research agent tasks curated from NLP, math, code, bio or cheminformatics, and time series. Tasks provide only  problem or dataset or metric specification, and agents must generate, debug, and execute training pipelines whose outputs are scored by fixed evaluation scripts. The paper defines a task configuration standard, run a large empirical study across two harnesses (AIRA-DOJO and MLGym) and multiple scaffolds/models, and propose an evaluation protocol combining valid submission rate, a normalized score transform, and Elo ratings fit via a BradleyTerry model.

**Audience:**

Yes

**Audience Explanation:**

1. The benchmark design is sound and valuable: fixed evaluation scripts, explicit metadata, and cross-harness compatibility reduce environment variance.
2. The “march of 9s” transform is conceptually appealing for reflecting distance-to-optimum.
3. The breadth of agent and scaffold configurations and multiple seeds is a strong point. Reporting VSR alongside quality metrics is insightful and aligns with reliability concerns.

**Claims And Evidence:**

Yes

**Claims Explanation:**

1. The “no-baseline-code” task design pushes agents to perform the full empirical ML workflow,  meaningful step beyond short-horizon or partial-pipeline benchmarks.
2. A task configuration standard that is convertible across harnesses (AIRADOJO to  MLGym) is useful and promotes cross-framework comparability.
3. Broad check of models and scaffolds, including sequential and tree or greedy exploration, with multiple seeds and long, fixed compute budgets, enables informative comparisons.
4. They report per-task distributions, difficulty stratification, and bootstrap CIs on Elo, which help interpret variability and headroom.

**Requested Changes:**

1. How do you quantify and control for retrieval of baseline implementations from the internet?
2. Why normalize with s_min as the worst observed score rather than a fixed trivial baseline?
3. What selection criteria reduced almost100 initial tasks to the final 20?